# Influence of Process Parameters in Three-Stage Purification of Aluminate Solution and Aluminum Hydroxide

Vladimir Damjanovic [1], Radislav Filipovic [1], Zoran Obrenovic [1], Mitar Perusic [2], Dusko Kostic [2], Slavko Smiljanic [2] and Srecko Stopic [3,*]

1 Alumina d.o.o., Karakaj, 75400 Zvornik, Bosnia and Herzegovina; vladimir.damjanovic@birac.ba (V.D.); radislav.filipovic@birac.ba (R.F.); zoran.obrenovic@birac.ba (Z.O.)

2 Faculty of Technology, University of East Sarajevo, Karakaj 34 A, 75400 Zvornik, Bosnia and Herzegovina; mitar.perusic@tfzv.ues.rs.ba (M.P.); dusko.kostic@tfzv.ues.rs.ba (D.K.); slavko.smiljanic@tfzv.ues.rs.ba (S.S.)

3 IME Process Metallurgy and Metal Recycling, RWTH Aachen University, Intzestrasse 3, 52056 Aachen, Germany

* Correspondence: sstopic@ime-aachen.de; Tel.: +49-176-7826-1674

**Abstract:** The influence of process parameters in the three-stage purification of aluminate solution from the Bayer process and aluminum hydroxide was considered in this paper. One of the ways of purification is treating the aluminate solution in order to reduce the concentrations in the starting raw material (solution) and then treating the aluminum hydroxide at a certain temperature and time in order to obtain an alumina precursor of adequate quality. The purification process itself is divided into three phases. The first phase involves the treatment of sodium aluminate with lime in order to primarily remove $Ca^{2+}$ and $(SiO_3)^{2-}$ impurities. Phase II aims to remove impurities of $Zn^{2+}$, $Fe^{2+}$, and $Cu^{2+}$ by treatment with controlled precipitation using specially prepared crystallization centers. In Phase III, $Na^+$ is removed by the process of hydrothermal washing of $Al_2O_3 \cdot 3H_2O$. In this work, parameters such as temperature (T), reaction time (t), and concentration of lime (c) were studied in order to remove the mentioned impurities and obtain the purest possible product that would be an adequate precursor for special types of alumina.

**Keywords:** Bayer process; sodium aluminate; alumina; hydrothermal process; aluminum hydroxide





## 1. Introduction

The Bayer process is the most important and widespread process for the production of alumina from bauxite, which is most often used for the production of metallic aluminum. By leaching the ore with sodium hydroxide at high temperatures and pressures, Bayer's solution of sodium aluminate is created. The solution has aluminum and sodium as the main components, but it also contains various impurities. Impurities adversely affect the stability of the aluminate solution and its properties and later affect the quality of the final product obtained from it (alumina and aluminum hydroxide). These impurities are bound into the crystal lattice of aluminum hydroxide, causing a change in its properties, which makes it difficult to use or even limits its use in certain final applications. In order to avoid the harmful effects of impurities, they must be removed by certain processes, i.e., their concentration must be lowered below the permitted limit. Special types of alumina (also identified as non-metallurgical alumina) are synthetic products obtained from bauxite ores in controlled processes. Depending on the applied process, it is possible to obtain different forms and types of alumina, namely, active (transitional), medium calcined, high calcined, tabular, and fused alumina. Each obtained product has characteristic properties, such as high mechanical strength and resistance to high temperatures and corrosion. Due to the stated properties, special alumina are the main components in various applications. They can be used as adsorbents, ceramic and refractory materials, abrasives, separation

membranes, semiconductors, then in bionics, orthopedics, electronics, military industry, etc. [1].

The most important and widespread process for the production of alumina is the Bayer process, which uses bauxite as raw material. There are four main stages of this process [1], namely, leaching, filtration, crystallization, and calcination, which lead to formation of aluminates at elevated temperatures. Sodium aluminate is an inorganic compound that is used as an efficient source of aluminum hydroxide in the Bayer process of producing alumina. Pure sodium aluminate (anhydrous) is a white crystalline solid that has the abbreviated formula $NaAlO_2$. Commercial sodium aluminate is available as a solution or as a solid [2]. The structure of the aluminate ion is shown in Figure 1.

**Figure 1.** Structure of the aluminate ion.

From the point of view of the production of alumina, the most important is sodium aluminate, which can be formed through the bauxite leaching reaction with NaOH solution and the bauxite sintering reaction with soda ($Na_2CO_3$) and subsequent dissolution of alumina using water [3]. Aluminate solutions are unstable and constantly hydrolyze with the formation of $Al(OH)_3$ and NaOH. The production of alumina according to the Bayer process is based on this important feature. The stability of aluminate solutions depends on several parameters, the most important of which are [4] concentration, caustic ratio, temperature, and content of impurities. Industrial aluminate solutions always contain impurities, which affect its stability as well as the quality of the obtained alumina. However, the impact of impurities is not sufficiently known, and their examination and removal is the subject of this paper. The focus of this work is the removal of calcium, silicon, iron, copper, and zinc from sodium aluminate solutions [5].

Impurities in aluminate solutions can be organic or inorganic. Organic impurities are mostly oxalates, acetates, bitumen, etc. Inorganic impurities represent dissolved elements ($Fe^{2+}$, $Zn^{2+}$, $Cu^{2+}$, $Ca^{2+}$, $Na^+$, $(SiO_3)^{2-}$, etc.). During leaching of bauxite at elevated temperatures and pressures at which aluminum dissolves from the ore, there is also a partial dissolution of impurities that pass into the solution [6–8]. In the used solution, the silicon comes from the aluminum hydroxide that was dissolved during the preparation of the synthetic solution. The effect of the presence of silicon on the hydroxide is due to adsorption from the solution during the precipitation of $Al(OH)_3$. It is dissolved in the form of sodium silicate ($Na_2SiO_3$), which is formed when dissolved in NaOH solution.

Dissolved iron in aluminate solutions is most often found in the form of ferrate ions ($Fe(OH)_4$), and it can also be found in the form of colloidal particles. Iron in bauxite is found in the mineral forms of goethite and hematite. Iron dissolution occurs during bauxite leaching [9].

Due to the instability of $Zn^{2+}$ ions in alkaline solutions, zinc is most often found in aluminate in the form of zincate ($ZnO_2^{2-}$ or $Zn(OH)_4^{2-}$). Colloidal zinc can also be formed, which, like any colloidal solution, cannot be separated by classical industrial filtration [10].

Silicon dioxide is commercially the most undesirable impurity of bauxite. This is because during the ore refining process in the Bayer process, insoluble sodium aluminosilicate is formed, which is separated from the process suspension as such, thus leading to the loss of the most important components (sodium hydroxide and aluminum oxide). Most bauxite ores contain silica in various forms. Forms of silicon dioxide that dissolve under given conditions are particularly relevant for the Bayer process, and these are most often clays and quartz. The silica that dissolves in the Bayer autoclave process is known as "reactive silica". Usually, the content of reactive silica is expressed as wt. % $SiO_2$. During leaching at low

temperatures (150 °C), the content of reactive silicon dioxide is proportional to the content of silicon dioxide present in the form of kaolinite, considering that kaolinite is completely soluble in Bayer suspension even at 150 °C. During leaching at high temperature (>240 °C), a part of the quartz is also decomposed, so the content of reactive silica will be equal to the sum of the silica present in the kaolinite and the decomposed quartz [6,11,12].

In order to know the behavior of calcium in Bayer's solutions, it is necessary to know the influence of the composition on the solubility of calcium. The presence of some organic species, such as humic acid and sodium gluconate, significantly increase the solubility of calcium because calcium forms soluble complexes with these organic species. In addition, high concentrations of carbonate have the effect of increasing the concentration of calcium in the solution, while phosphate reduces solubility at low concentrations of carbonate. Initially, this behavior was attributed to the higher solubility of calcium carbonate compared to tricalcium aluminate ($3CaO \cdot Al_2O_3$; TCA). However, according to Le Chatelier's principle, calcium concentration should decrease with increasing carbonate concentration. The previous observations can be explained by the fact that the most unstable species are also the most soluble, that is, the main soluble species in Bayer suspensions is the monomer calcium aluminate, in equilibrium with hemicarbonate species $C_4A$ ($4CaO \cdot Al_2O_3$) [13].

Regarding purification of aluminate solutions, an experimental work was performed using standard glassware, A.R. chemicals, instruments, and methods at the research laboratory at "Alumina DOO. Zvornik, Bosnia and Herzegovina [14]. In this work, it was possible to remove iron, zinc, and copper from Bayer liquor with an efficiency of more than 90% in such a way that the treated solution is still economically usable in the following stages of processing while obtaining different types of aluminum trihydrate. Based on the results presented, it can be firstly concluded that an increasing time of contact between the liquor and seed crystals has a positive effect on the removal of iron, zinc, and copper liquor impurities. Increasing the temperature indirectly reduces the impurity removal efficiency from the sodium aluminate solution because higher process temperature results in the rate of precipitation being lower and the solubility of all impurities being higher. At a temperature of 40 °C, good impurity removal results are achieved.

A Bayer process solution is filtered through a bed of particles of a granular substance containing $Fe_2O_3$ to remove copper and zinc species from the solution. The particles preferably have a $Fe_2O_3$ content of about 40 to 100 wt. %. For more effective removal of zinc, the particles are coated with a metal sulfide, preferably zinc sulfide [15]. A maximum zinc content of 0.03 wt. % can be tolerated in certain alloys. Theoretically, a zinc content of 0.03 wt. % in the metal is derived from an aluminum oxide containing about 0.02 wt. % ZnO. To produce satisfactory aluminum metal, the CuO and ZnO contents of aluminum oxide should generally be held to less than about 0.015 and 0.023 wt. %, respectively.

Removal of some impurities from Bayer liquor, such as zinc compounds, allows alumina with low content of impurities incorporated in crystalline structure to be obtained [16,17]. Milovanovic et al. [18] studied the purification of Bayer liquor. Crystallization of Bayer liquor was conducted at 52 °C for 24 h, whereas aluminum hydroxide with specific structural properties was used as seed. The crystallization product (aluminum hydroxide) was calcined at 950 °C for 2 h with a heating rate of 5 °C/min. The obtained alumina (Alumina I) was compared with alumina obtained without Bayer liquor purification (alumina from bauxite refinery "Alumina" d.o.o Zvornik, Bosnia and Herzegovina (Alumina II). Zinc as zinc oxide in the initial and purified Bayer liquor was 0.0494 L and 0.0057 g/L, respectively. Alumina from the bauxite refinery contained 0.026 wt. % ZnO, whereas the zinc content in alumina obtained after Bayer liquor purification was 0.016%.

Dissolution of iron occurs during digestion, which represents an equilibrium concentration established between the iron-bearing minerals in the bauxite and the liquor. Ibrahim et al. [19] mentioned that iron is present as colloidal or nanosized particles rather than as a solution and passes through security filtration. The hematite surface area, lime addition, free caustic concentration, temperature, holding time, and mineralogy all influence the liquor iron concentration. Elevated $SO_4$, $CO_3$, and F concentrations in pregnant green liquor

(PGL) would also appear to favor lower iron liquors. As is evident from plant data, lime injection into the digesters at 270 °C can bring down the iron-in-liquor concentration by up to 7 mg/L. Lime addition is an important process for iron control and helps to dramatically overcome high iron concentrations. To target a lower iron-in-liquor concentration, an incremental approach is required, and for this reason, a range of small improvements need to be targeted simultaneously. Holding time and addition of reagents can help with the growth of colloidal iron particles, while an increased colloid size will aid its removal when using flocculants and during filtration [19].

This work was focused on the purification of aluminate solution using an improved research strategy containing three phases. There are many ways to remove impurities depending on the impurities involved. For example, organic dissolved substances are removed by adding whitening additives, which coagulate dissolved organic compounds.

In this work, the focus was on the removal of impurities of iron, zinc, silicon, calcium, and sodium. Some of the ways to achieve this include removal of $Zn^{2+}$ and $Cu^{2+}$ by filtration through a layer of granules containing iron trioxide [10], removal of $Zn^{2+}$ by addition of ZnS germs in the presence of sulfide ion [13], removal of colloidal iron by filtration through a suitable polymer [13], and removal of $Fe^{2+}$ by filtration through a layer of sand [9]. The main aim of this work was to remove the mentioned impurities from synthetic sodium aluminate, which was prepared from a non-metallurgical hydrate that dissolves in sodium hydroxide. The sodium hydroxide solution was prepared by dissolving granulated solid NaOH in a certain amount of water.

## 2. Materials and Methods

### 2.1. Material

Non-metallurgical hydrate obtained at the factory "Alumina d.o.o", Zvornik, Bosnia and Herzegovina, was used as raw materials to obtain ultrapure aluminum hydroxide and pure alkali (base), lime ($CaO_{akt}$ = 90.93%), and finely precipitated aluminum hydroxide.

### 2.2. Characterization of the Liquid Phase

Chemical analysis of elements dissolved in the solution was obtained by high-resolution inductively coupled plasma optical emission spectroscopy (ICP-OES) which allows the determination of elements with an atomic mass number range of 7 to 250 (Li to U). The equipment used was a "Spectro Genesis" spectrometer.

### 2.3. Characterization of the Solid Phase

The chemical composition of the sample was determined by fluorescence X-ray structural spectroscopy (EDX) on a Shimadzu 8000 device. Bulk analysis, especially for impurities, was carried out by AAS or ICP, after the purification step and shown in Appendix A of the text.

All samples were analyzed under the following conditions: beam voltage of 10 kV, analyzed X-ray energy range of 0 to 10 keV, beam frequency of 20,000 pulses per second, time acquisitions of 300 s. Composition identification was performed using DIFFRAC.SUITE EVA FT software, Version 4.2. IR spectra were recorded using Shimadzu IRAffinity-1S (single reflection) infrared spectrophotometry with Fourier transform. The analysis was performed in the range of wavenumbers 4000–400 $cm^{-1}$, with 20 scans and a resolution of 4 $cm^{-1}$.

The SEM analysis was performed on the JSM 7000F by JEOL (construction year 2006, JEOL Ltd., Tokyo, Japan) and EDX analysis was performed using the Octane Plus-A by Ametek-EDAX (construction year, 2015, AMETEK Inc., Berwyn, PA, USA) with software Genesis V 6.53 by Ametek-EDAX, revealing an irregular structure of solid residue.

XRD analysis of solid residue in the first purification phase was performed using a Bruker D8 Advance with a LynxEye detector (Bruker AXS, Karlsruhe, Germany). X-ray powder diffraction patterns were collected on a Bruker AXS D4 Endeavor diffractometer

in Bragg–Brentano geometry, equipped with a copper tube and a primary nickel filter providing Cu K$\alpha$1,2 radiation ($\lambda$ = 1.54187 Å).

*2.4. Procedure*

In order to obtain the highest quality product (aluminum hydroxide), the aluminate solution must be purified from the various impurities contained in it. Depending on the temperature and type of bauxite, pressure, as well as other parameters, a smaller or larger amount of various organic and inorganic impurities were dissolved in sodium aluminate.

Empirically, it was determined that 204 g of solid NaOH per liter of demineralized water is needed to dissolve 260 g of non-metallurgical hydrate. In this way, an aluminate solution with 170 and 155 g/L concentrations of $Al_2O_3$ and $Na_2O$ was obtained, which then underwent purification.

Purification of the mentioned impurities was carried out in three stages, as shown in Figure 2:

- Phase I: Addition of lime in order to remove calcium and silicon.
- Phase II: Rapid crystallization by adding aluminum hydroxide as a crystallization initiator in order to remove iron, zinc, and copper impurities.
- Phase III: Washing in autoclaves to remove sodium impurities.

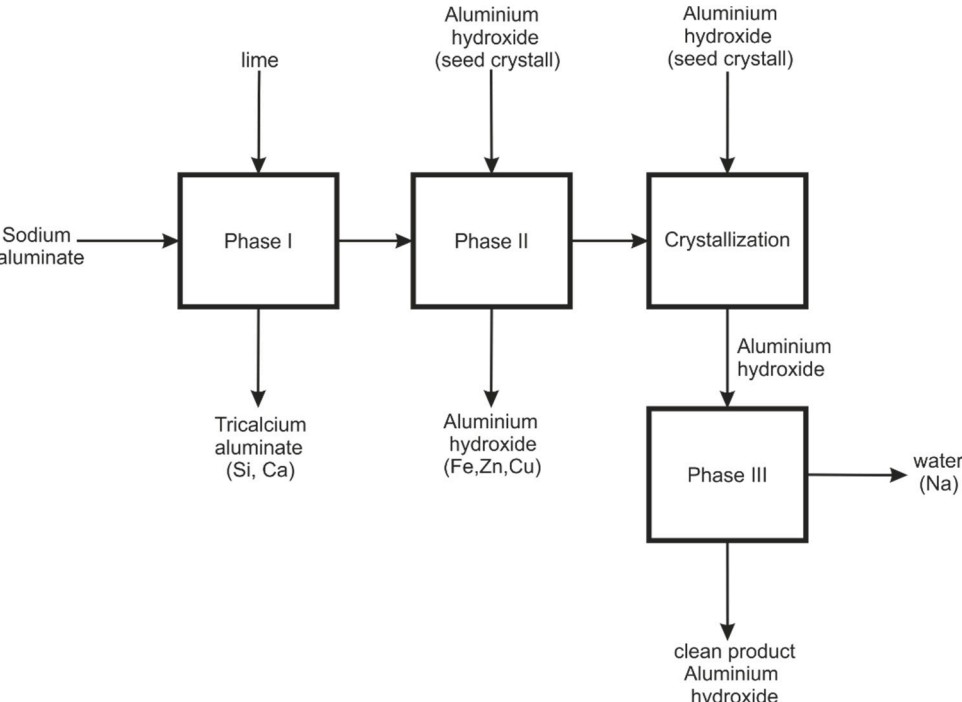

**Figure 2.** Schematic representation of the process.

The influence of time, lime concentration (Phase I), amount of crystallization initiator (Phase II), hydrothermal washing of aluminum hydroxide (Phase III), and treatment temperature was monitored. Each of the parameters was monitored with constant values of the other two parameters. Sodium aluminate was obtained by dissolving non-metallurgical aluminum hydroxide. Non-metallurgical aluminum hydroxide was produced at the factory and delivered to the laboratory. At an elevated temperature, aluminum hydroxide was treated with sodium hydroxide solution, resulting in the so-called synthetic aluminate. The concentration of the obtained solution and the caustic modulus were adjusted to be at the level of the process Bayer liquor. Aluminate was also analyzed on the ICP, where the impurity content was determined before treatment with the appropriate reagent. In contrast to the first purification phase, removal of $Fe^{2+}$, $Zn^{2+}$, and $Cu^{2+}$ in the second phase

had a physical nature based on some adsorption mechanisms on very fine powder of added aluminum hydroxide.

## 3. Results and Discussion

### 3.1. First Step of the Purification Process

As already mentioned, the first phase involved the purification of the aluminate solution from calcium ions and silicon ions. The following chemical reaction are expected in the first purification step.

$$Al_2O_3 + 2NaOH \rightarrow 2NaAlO_2\ (aq) + 2H_2O \tag{1}$$

$$SiO_2 + 2NaOH \rightarrow Na_2SiO_{3\ (aq)} + H_2O \tag{2}$$

$$2NaAlO_2 + Ca(OH)_2 \rightarrow Ca_3Al_2O_{6\ (s)} + 2NaOH \tag{3}$$

$$Na_2SiO_3 + Ca(OH)_2 \rightarrow CaSiO_3\ (s) + 2NaOH \tag{4}$$

$$3Al_2Si_2O_5(OH)_4\ (s) + 18NaOH \rightarrow 6Na_2SiO_3 + 6NaAl(OH)_4 + 3H_2O \tag{5}$$

$$6Na2SiO_3 + 6NaAl(OH)_4 + NaX \rightarrow Na_6[Al_6Si_6O_{24}] \cdot 2\ NaX\ (s) + 12NaOH + 6H_2O \tag{6}$$

The process was carried out by adding lime to sodium aluminate and monitoring the effects of the process parameters (time, temperature, and concentration of lime) Table 1. After the experiments were completed, filtration was performed, where the phases were separated into liquid and solid. The liquid phase (filtrate) that had been purified from silicon and calcium ions was subjected to further purification (Phase II).

**Table 1.** Process parameters for the first stage of the process.

| Synthesis No. | Time of Synthesis (min) | Temperature (°C) | Concentration of Lime (g/L) |
|---|---|---|---|
| 1A | 60 | 80 | 20 |
| 2A | 120 | 80 | 20 |
| 3A | 180 | 80 | 20 |
| 4A | 240 | 80 | 20 |
| 5A | 300 | 80 | 20 |
| 6A | 120 | 50 | 20 |
| 7A | 120 | 60 | 20 |
| 8A | 120 | 70 | 20 |
| 9A | 120 | 80 | 20 |
| 10A | 120 | 80 | 5 |
| 11A | 120 | 80 | 10 |
| 12A | 120 | 8 | 20 |
| 13A | 120 | 80 | 30 |
| 14A | 120 | 80 | 40 |

To obtain a volume of 1 L of synthetic aluminate with caustic ratio $\alpha_k = 1.50$ (tolerance 1.45–1.55), 204 g of sodium hydroxide and 260 g of dry nonmetallurgical hydrate were weighed on an analytical balance. The measured NaOH was transferred to a 2 L beaker, and

water was then added up to 350 mL. This solution was first heated in order to completely clarify it, and dry nonmetallurgical hydrate was then moderately added to it. The resulting solution was still heated to boiling temperature and allowed to boil for 15 min. After that, the resulting aluminate solution was cooled to room temperature and transferred to a 1 L flask, with water added to the line. Then, 100 mL of the solution was taken to determine the content of $Al^{3+}$, $Na^+$, $Ca^{2+}$, $(SiO_3)^{2-}$, $Zn^{2+}$, $Cu^{2+}$, and $Fe^{2+}$ in the initial solution, while the rest was transferred to a beaker and heated to the synthesis temperature. A given amount of lime was added to the heated solution under the synthesis conditions.

### 3.1.1. The Influence of Time

The influence of time on the purification of silicon and calcium was measured at a temperature of t = 80 °C by adding an additional 20 g/L of lime at different times ranging from 60 to 300 min. Figure 3 shows the recorded results.

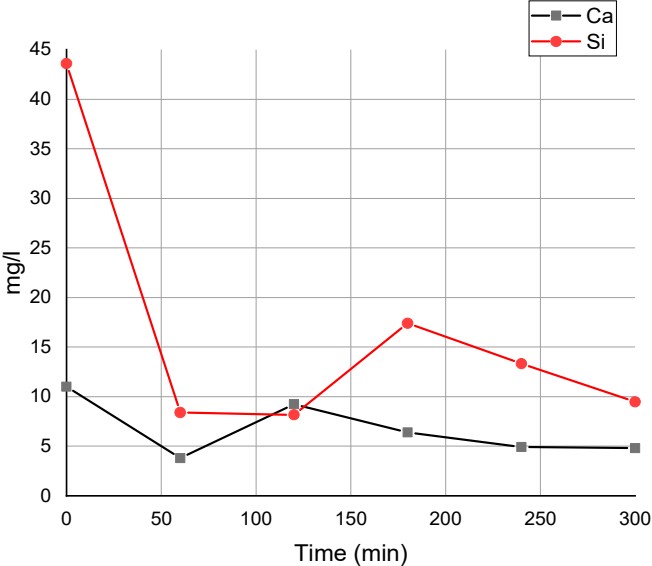

**Figure 3.** Change in Ca and Si concentrations over time (T = 80 °C, 60–300 min).

The content of Si first decreases, then increases and decreases again over time, and has the lowest value at 120 min, while the content of Ca is the lowest after 60 min and its value is constant but lower compared to the initial one. The reason for this is the formation of insoluble tricalcium—aluminate ($Ca_3Al_2O_6$). It is clearly visible from Figure 3 that at a temperature of 80 °C and an amount of lime of 20 g/L, the time after which the process should be stopped is 60 min.

### 3.1.2. Effect of Temperature

Figure 4 shows the effect of temperature on the content of Si and Ca in the aluminate solution after the addition of 20 g/L of lime and 120 min of reaction time.

At temperatures below 50 °C, calcium removal was very slow. It can be seen from the previous diagram that the concentration of calcium was the lowest at 50 °C, where its concentration was 4.16 mg/L. With a further increase in temperature, the concentration of Ca increased, which can be explained by the fact that calcium compounds dissolve in sodium aluminate solution at higher temperatures. However, all values were lower compared to the initial value of 10.64 mg/L. Unlike calcium, the purification of silicon was more intense at higher temperatures, while the minimum concentration was at 70 °C. All values were well below the initial value of 43.88 mg/L. One of the mechanisms for extracting silicon is the formation of DSP (desilication product), which is actually a compound of sodium aluminosilicate (a type of zeolite sodalite). This process represents a loss of sodium and aluminum in conventional processes. Our new innovative strategy in this work is the multistep cleaning of the aluminate solution with special reference to the removal

of impurities from the second stage. It is known from the literature that the kinetics of sodium aluminosilicate (DSP) formation as well as crystallinity increase with increasing temperature, as shown in [6]:

$$6Na_2SiO_3(aq) + 6NaAl(OH)_4(aq) + Na_2X(aq) \rightarrow Na_6(Al_6Si_6O_{24}) \cdot Na_2X \cdot 3H_2O(s) + 12NaOH\ (aq) + 3H_2O\ (aq)$$

$$X = CO_3^{2-}, SO_4^{2-}, 2OH^-, 2Cl$$

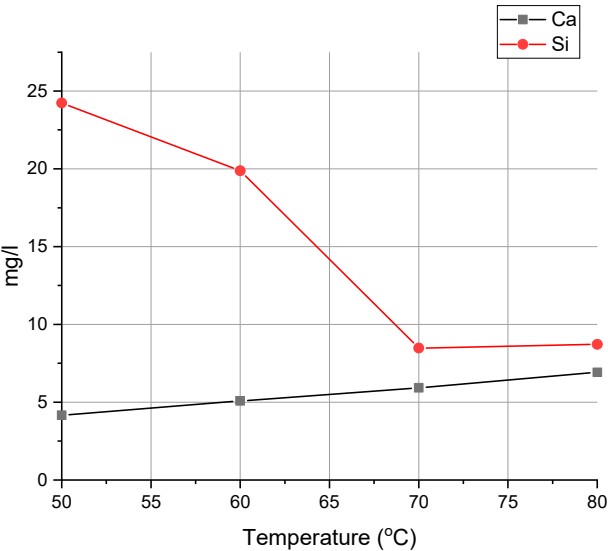

**Figure 4.** Change in Ca and Si concentrations for different temperatures (120 min, T = 50–80 °C).

This process was most effective at temperatures of 90–100 °C; however, high temperatures were not favorable due to the increase in calcium in the solution. Due to the presence of calcium in excess, there was an exchange of calcium ions with sodium at the cation position in the aluminosilicate structure, reducing the loss of NaOH due to the formation of calcium aluminosilicate cancrinite. This is another mechanism for extracting calcium from the solution. Also, considering the cage structure of DSP (sodalite), which is also shown by the previous equation, carbonate and sulfate ions were present. Considering the addition of lime and the resulting reaction, it is very likely that calcium carbonate was present.

### 3.1.3. Effect of Lime Concentration

The influence of the concentration of added lime was determined at the following process parameters: T = 80 °C t = 120 min, and concentration of lime = 5–40 g/L. Even with different concentrations of added lime, the amount of sodium present in the solution did not change and remained the same as in the initial sample. As the amount of added lime increased, the content of $Al_2O_3$ in the solution decreased. By increasing the concentration of calcium present, there was a more intense reaction with aluminum from the aluminate, which resulted in the formation of insoluble tricalcium aluminate. The assumption is that this phenomenon occurs by two possible mechanisms.

One is that as a result of oversaturation of the solution with CaO oxide, a very fast reaction of formation of highly soluble tetracalcium aluminate occurs, which eventually transforms into hardly soluble tricalcium aluminate, which precipitates very quickly and is also the center of crystallization. At the same time, the synthesis of TCA is additionally accelerated, which ultimately leads to a decrease in the solubility of calcium in the given conditions (the given conditions favor the precipitation of TCA below the solubility limit of CaO). We suggest that the transition compound tetracalcium aluminate, which is highly soluble, pulls all the calcium out of the solution in the synthesis, and there is then a transformation and precipitation of TCA and $CaCO_3$ from the solution.

Another mechanism that we assume takes place is the adsorption process. According to the methods used, soluble calcium means the entire amount of Ca that is detected via ICP-OES in the solution after filtering suspended particles on a laboratory filter. However, on these filters, it is not possible to separate colloidal particles of calcium compounds that end up in the solution. Tetracalcium aluminate is an intermediate compound. During transformation, TCA forms colloidal particles that are adsorbed on $Ca(OH)_2$ particles, which also has limited solubility in sodium aluminate solution. In this way, by the mechanism of adsorption of nanosized particles on the formed crystals of TCA and insoluble $Ca(OH)_2$, it is possible to separate them from the solution.

Figure 5 shows the dependence of the content of Si and Ca in the solution on the amount of added lime. The addition of relatively small amounts of lime is advantageous because there is a distinct removal of silicon in the solution. Adding 20 g/L of lime removed the most calcium. With the further addition of lime, the silicon value constantly decreased, while the calcium content in the solution increased slightly. After 120 min reactions and at a temperature of 80 °C, the optimal amount of added lime was adopted, i.e., 20 g/L. With the addition of 50 g/L of lime, the most silicon was removed due to the formation of insoluble hydrogarnet; however, the content of $Al_2O_3$ decreased. Therefore, the process can be carried out with the addition of 20 g/L of lime, whereby larger amounts of calcium are removed with significantly lower losses of aluminum.

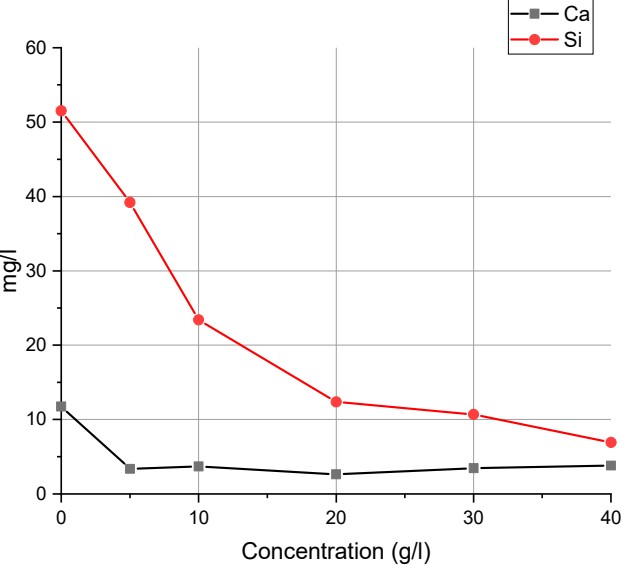

**Figure 5.** Influence of lime concentration on the change in Si and Ca content (T = 80 °C, 120 min, c = 5–40 g/L).

Figure 6 shows the infrared spectrum of the solid residue at a temperature of 80 °C, time of 120 min, and lime concentration of 20 g/L.

At the wavelength of 805 $cm^{-1}$, a medium-sized signal was observed, which originated from the symmetric stretching of the Si-O-Si bond in hydrogarnet ($Ca_3Al_2(SiO_4)_3 - x(OH)_{4x}$). This confirmed the presence of silicon compounds that were not present in the starting lime. The calcium compound hydrogarnet ($CaO·Al_2O_3·6H_2O$) was characterized by an absorption band at the wavelength of 540 $cm^{-1}$, which originated from the vibrations of the Ca-O bond. The absorption band in the infrared spectrum that appeared at the wavelength of 3660 $cm^{-1}$ was due to the stretching of O-H bonds.

Additional morphological and structural analysis were performed using SEM, EDX, and XRD analysis, as shown in Figures 7–9. The obtained particles agglomerated in irregular and round forms, with particle size being above 1 μm.

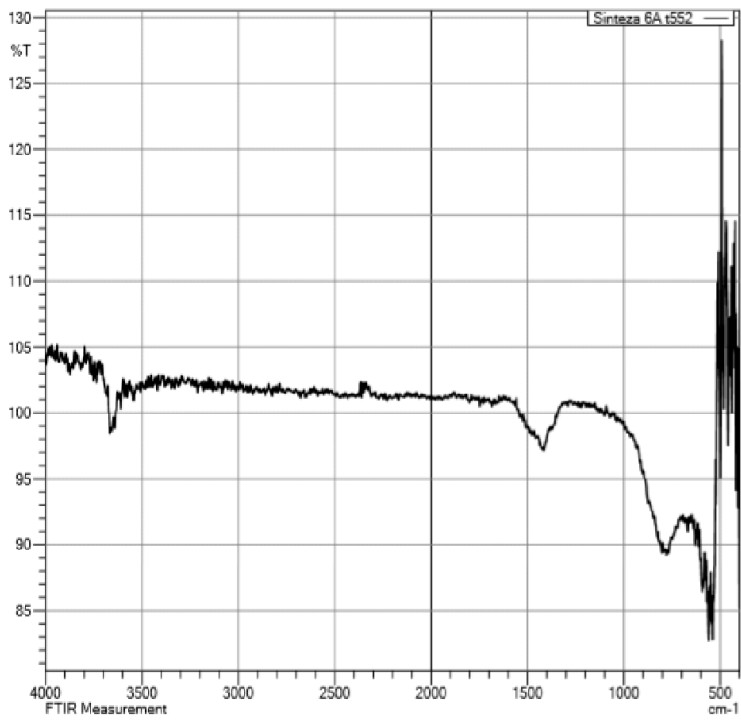

**Figure 6.** Infrared spectrum of the solid residue (T = 80 °C, 120 min, 20 g/L).

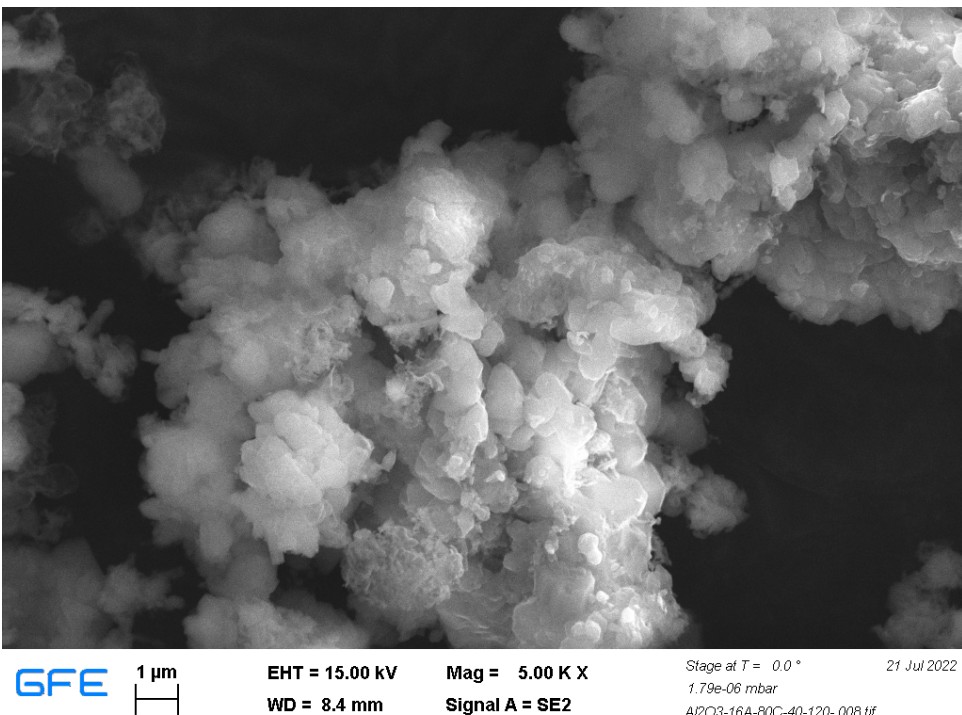

**Figure 7.** Typical SEM analysis of solid residue after the first purification step (14A: T = 80 °C, 120 min, 40 g/L).

EDX analysis revealed the presence of residual Ca and Si. Generally, the presence of calcium and silicon was confirmed at different places of powders. The structural analysis of powder was studied using XRD analysis, as shown in Figure 9.

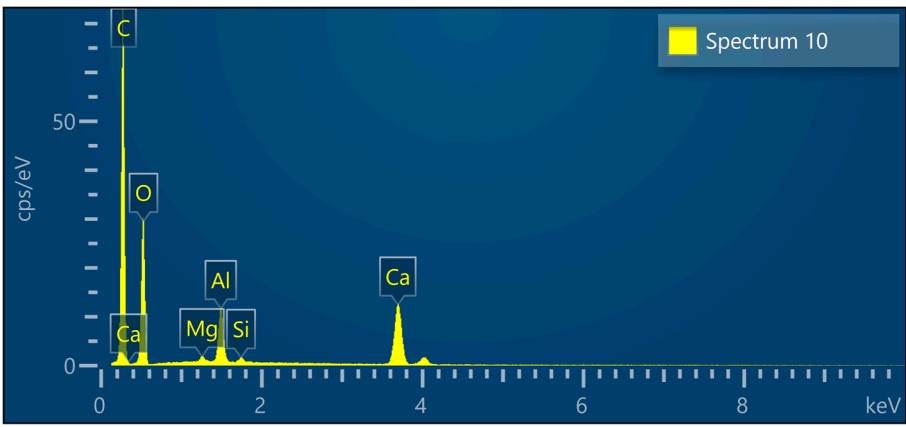

**Figure 8.** EDX analysis of solid residue (14A: T = 80 °C, 120 min, 40 g/L).

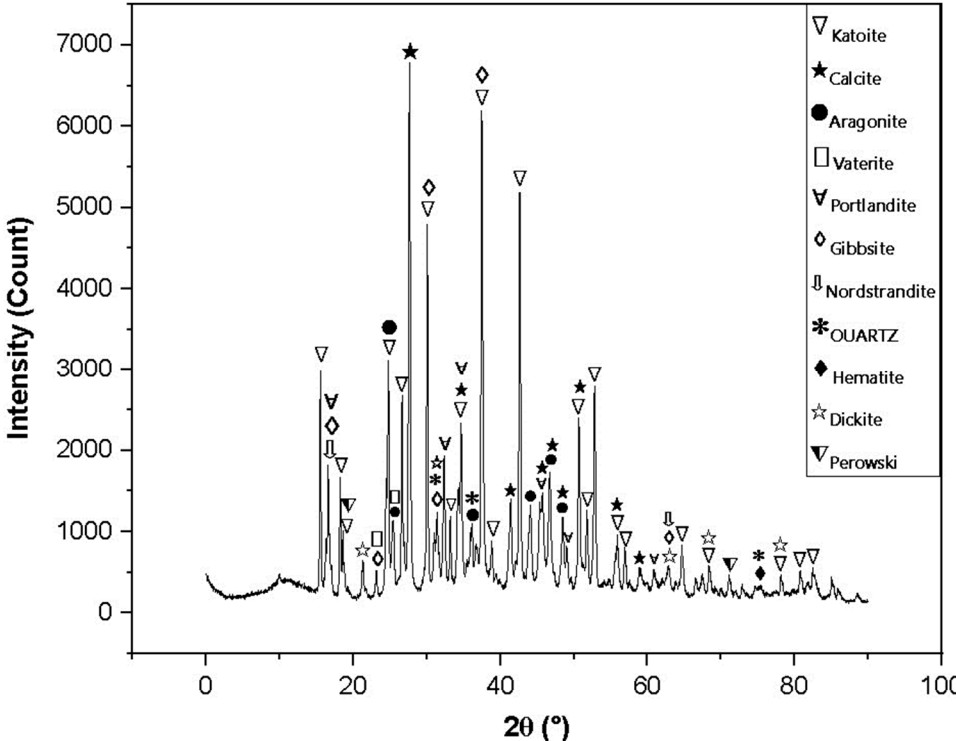

**Figure 9.** XRD analysis of solid residue (14A: T = 80 °C, 120 min, 40 g/L).

The sample analyzed contained predominantly a Ca-carrying hydrogarnet ($Ca_3Al_2(OH)_{12}$). The other main phases were calcite, aragonite, and gibbsite. Portlandite and vaterite were minor constituents (<10%). The wide peak at 2Θ = 10–16° indicated the presence of an amorphous phase, possibly amorphous $Ca(OH)_2$. Other minor phases were nordstrandite, dickite, quartz, perovskite, and possibly rhomboklas. There was little hematite and goethite. Cristobalite might be another minor constituent in the analyzed sample.

### 3.2. Second Step of the Purification Process

In the second phase of this work, the purification of the aluminate solution continued in order to obtain the purest possible product (aluminum hydroxide). Sodium aluminate from Phase I was further treated with very fine aluminum hydroxide (seed crystals), during which iron, zinc, and copper ions were adsorbed on the surface of the solid phase. This is an innovative step of this work, which is missing in the literature on purification of solutions [20–25]. Seed crystals were prepared after grinding of water suspension with very fine aluminum hydroxide. During the cleaning process, the sodium aluminate solution

broke down (crystallization), and the aluminum oxide content decreased over time. The purified solution was decomposed (crystallized), and the obtained product was sent to the third stage of the process. The solution was prepared in the same way as in the previous phase, with the difference being that in this phase, instead of lime, specially prepared aluminum hydroxide was added.

### 3.2.1. The Influence of Time

As shown in Table 2 (blue color), the cleaning of the aluminate solution was monitored for different times, i.e., the change in the concentration of $Fe^{2+}$, $Zn^{2+}$, and $Cu^{2+}$ with the other parameters (temperature and seed crystal concentration) remaining constant, as shown in Figure 10.

**Table 2.** Process parameters for the second stage of the process.

| Synthesis No. | Time of Synthesis (min) | Temperature (°C) | Concentration of Seed Crystal (5 g/L) |
|---|---|---|---|
| 1B | 30 | 50 | 5 |
| 2B | 45 | 50 | 5 |
| 3B | 60 | 50 | 5 |
| 4B | 120 | 50 | 5 |
| 5B | 150 | 50 | 5 |
| 6B | 120 | 45 | 5 |
| 7B | 120 | 50 | 5 |
| 8B | 120 | 60 | 5 |
| 9B | 120 | 70 | 5 |
| 10B | 120 | 50 | 5 |
| 11B | 120 | 50 | 10 |
| 12B | 120 | 50 | 20 |
| 13B | 120 | 50 | 30 |
| 14B | 120 | 50 | 40 |

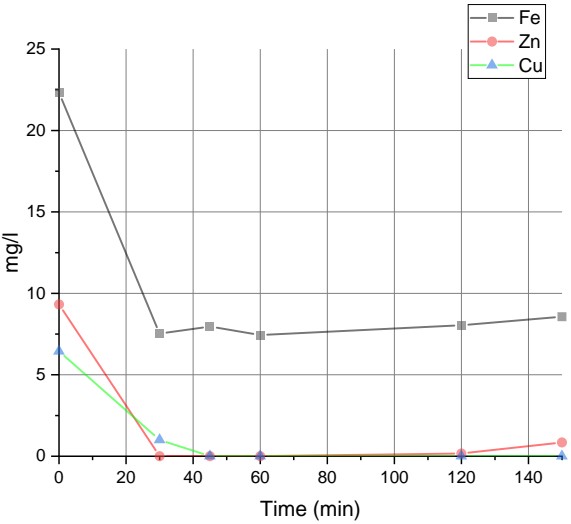

**Figure 10.** Changes in $Fe^{2+}$, $Zn^{2+}$, and $Cu^{2+}$ concentrations over time (T = 50 °C, c = 5 g/L, t = 30–150 min).

The results showed a significant decrease in the concentration of iron, zinc, and copper over time compared to the initial value. The concentration of copper already dropped to

zero after 40 min. The concentration of iron had the lowest value after 30 min, and the concentration then started to increase, which can be explained by desorption or dissolution from the surface of the seed crystals. The concentration of zinc was already equal to 0 after 30 min, but after 120 min, its concentration started to increase because of the same redissolution process. Therefore, the time of 45 min can be considered as optimal for the best purification effect.

### 3.2.2. Effect of Temperature

As in the previous phases, at a constant time of 120 min and a constant seed crystal concentration of 5 g/L, the influence of temperature on the cleaning of the aluminate solution was monitored and is shown in Figure 11.

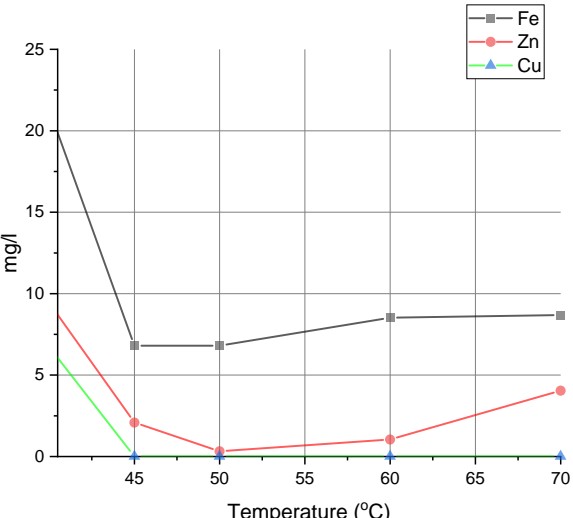

**Figure 11.** Changes in $Fe^{2+}$, $Zn^{2+}$, and $Cu^{2+}$ concentrations depending on temperature (120 min, 5 g/L).

It can be concluded that the influence of temperature on the concentration of impurities is specific, with the temperature of 50 °C giving the best results. As far as individual concentrations are concerned, copper is completely removed at all temperatures, iron increases, and zinc concentration first decreases and then increases with the increase in temperature.

### 3.2.3. The Influence of Seed Crystal Concentration

Finally, the influence of the concentration of specially produced aluminum hydroxide (seed crystal) on the process of cleaning sodium aluminate from the mentioned impurities was monitored. In this case, the seed crystal ratio changed, while the time (120 min) and temperature (50 °C) remained constant, as shown in Figure 12.

Based on the results from Figure 12, it can be clearly seen that the seed crystal concentration of 10 g/L completely eliminated all the observed impurities, which represents an ideal solution for this purification phase. As for the behavior of individual elements, the concentrations of copper and zinc decreased and the concentration of iron increased with the increase in the seed crystal concentration. It is important to note that an increase in the seed crystal concentration also increased the degree of decomposition of the aluminate solution, which was unfavorable and led to the loss of aluminum. Generally, the results shown on Figures 10–12 were obtained in repeated experiments, confirming that the proposed mechanism was well chosen, which is of high importance for the purification process.

SEM analysis of solid residue from the second purification step is shown in Figure 13.

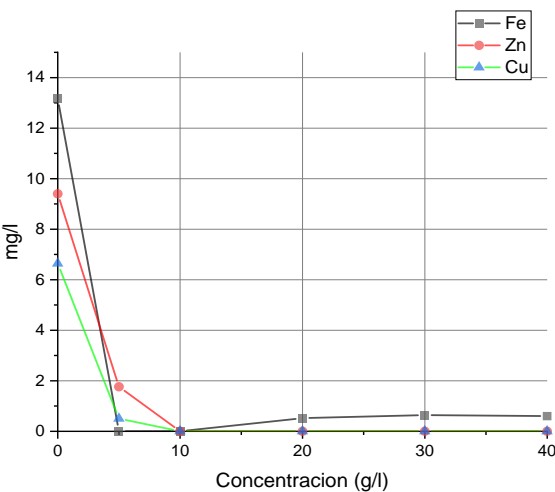

**Figure 12.** Influence of seed crystal concentration on changes in $Fe^{2+}$, $Zn^{2+}$, and $Cu^{2+}$ concentration (120 min, 50 °C, c = 5–40 min).

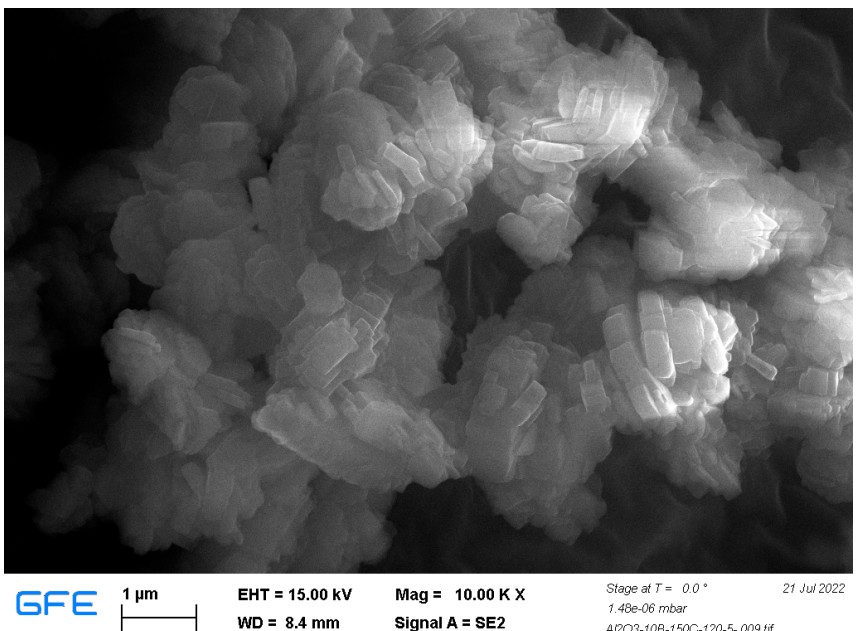

**Figure 13.** SEM analysis of powders after Phase II (10B: 120 min, 50 °C, 5 g/L).

The obtained particles were agglomerated and irregular, with particle size above 1 μm. EDX analysis of the solid residue after the second purification process is presented in Figure 14.

The obtained results confirmed the presence of sodium and aluminum in the solid residue. The removal of sodium was performed in the third step. Carbon was used as a carrier in the analysis.

### 3.3. Third Step of the Purification Process

The last stage of this process involved the purification of solid aluminum hydroxide by hydrothermal washing under different conditions, as shown in Table 3. The aim of hydrothermal washing (HTW) is to remove the sodium oxide present in the aluminum hydroxide. Taking into account all three stages of the process, a product with a reduced content of harmful impurities was obtained. It should be noted that during the hydrothermal washing process, a phase transformation from trihydrate to monohydrate occurred.

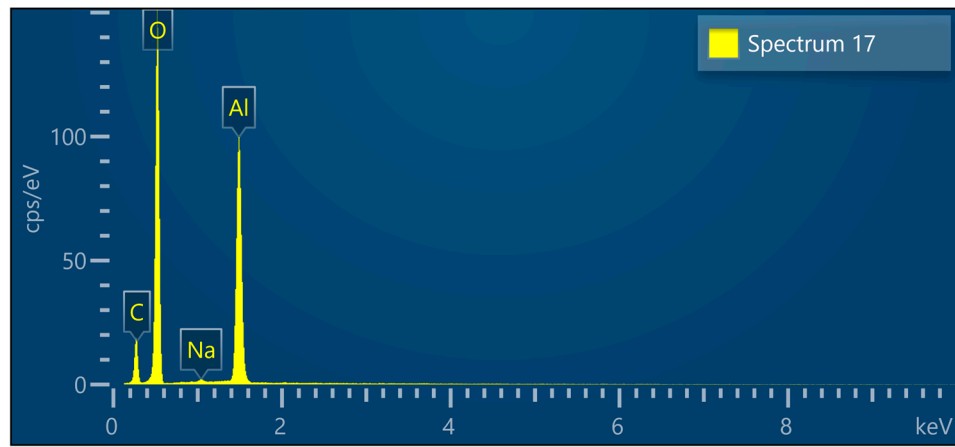

**Figure 14.** EDX analysis of the solid residue after the second purification step (10B: 120 min, 50 °C, 5 g/L).

**Table 3.** Process parameters for the third stage of the purification.

| Hydrothermal Washing No. | Time (min) | Temperature (°C) | Concentration of Hydrate (g/mL) |
| --- | --- | --- | --- |
| 1C | 30 | 210 | 167 g/L |
| 2C | 45 | 210 | 167 g/L |
| 3C | 60 | 210 | 167 g/L |
| 5C | 150 | 210 | 167 g/L |
| 6C | 120 | 170 | 167 g/L |
| 7C | 120 | 190 | 167 g/L |
| 8C | 120 | 200 | 167 g/L |
| 9C | 120 | 210 | 167 g/L |
| 10C | 120 | 230 | 167 g/L |
| 11C | 120 | 250 | 167 g/L |
| 12C | 120 | 210 | 16,7 g/L |
| 13C | 120 | 210 | 83 g/L |
| 14C | 120 | 210 | 167 g/L |
| 15C | 120 | 210 | 334 g/L |
| 16C | 120 | 210 | 500 g/L |

### 3.3.1. The Influence of Time

As in the previous phases, the washing time was first monitored with the other parameters (temperature and hydrate concentration) remaining constant, as shown in Figure 15.

Based on the data obtained in the experiment, it can be concluded that sodium concentration decreases over time and that reaches a minimum value at 150 min. However, the cleaning effect is very effective even for 60 min.

### 3.3.2. Effect of Temperature

In the second set of experiments, as in the previous phases, the influence of temperature on the purification of aluminum hydroxide from sodium oxide was monitored with the other parameters remaining constant (time of 120 min and hydrate concentration of 167 g/L), as shown in Figure 16.

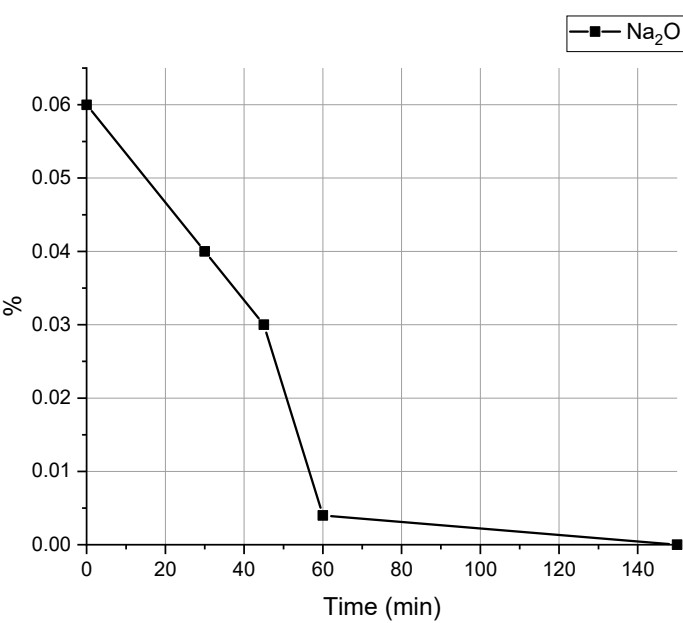

**Figure 15.** Effect of time on the reduction of sodium oxide concentration (210 °C, hydrate concentration of 167 g/L).

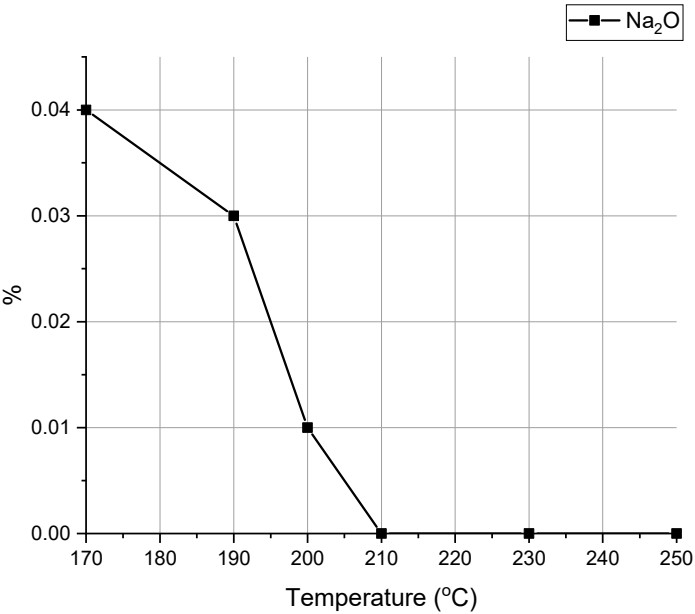

**Figure 16.** Effect of temperature on the reduction of sodium oxide concentration (120 min, hydrate concentration of 167 g/L, temperature between 170 and 210 °C).

It can be seen from the graph that as the temperature increased, the degree of purification of the product also increased. The lowest purification effect was at the temperature of 170 °C, and the purification was already complete at the temperature of 210 °C, i.e., the share of sodium oxide dropped to 0%. As expected, a further increase in temperature achieved the same effect as the given temperature of 210 °C.

### 3.3.3. Effect of Hydrate Concentration

In the third set of experiments, the influence of hydrate concentration on purification was monitored with constant values of temperature of 210 °C and time of 120 min, as shown in Figure 17.

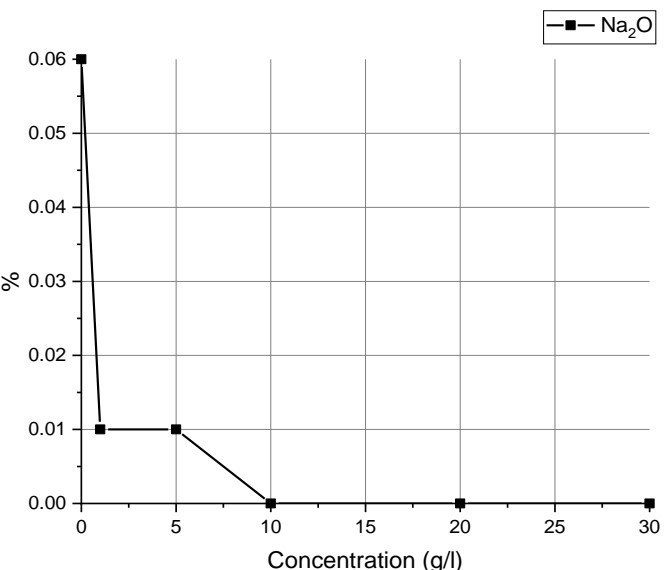

**Figure 17.** Effect of hydrate concentration on the reduction of sodium oxide content (210 °C, 120 min).

Based on the graph in Figure 17, it can be seen that the concentration of sodium oxide already dropped to a value of 0.01% at a hydrate concentration of 16.7 g/L. Further increase in the hydrate concentration led to a value of 0% at a used ratio of 167 g/L. Further increasing the concentration had no effect because all the sodium had been removed even at lower values, which was our final aim.

The typical XRD analysis of the cleaned solid material is shown in Figure 18.

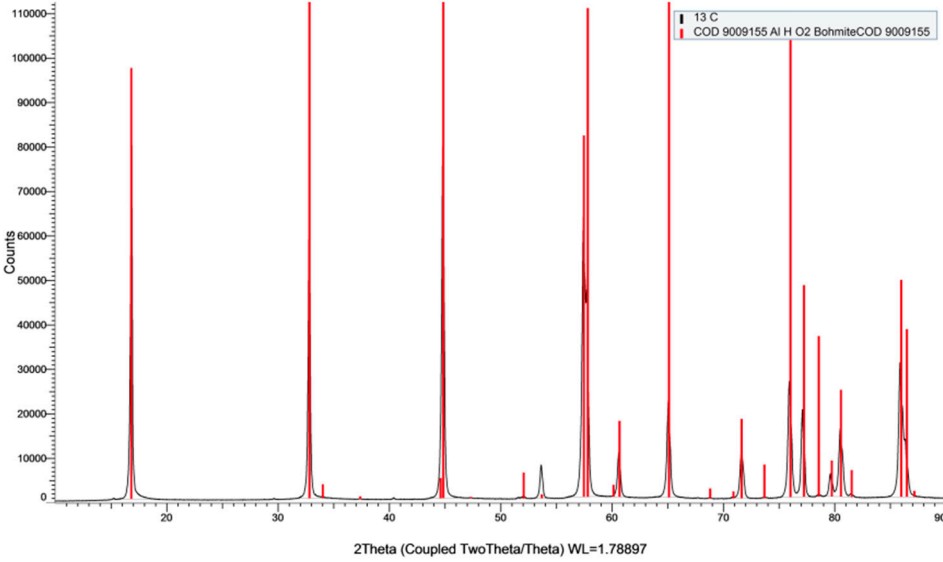

**Figure 18.** XRD analysis of solid product in the third purification step (210 °C, 120 min, 83 g/L).

The structure of boehmite (AlOOH) was detected in our solid residue from the third purification step, which was formed from gibbsite at 210 °C.

## 4. Conclusions

A review of the literature in the last 30 years concluded that the current methods of purifying sodium aluminate are not efficient enough or are too complex and expensive. Generally, this purification process is very important for the production of alumina with the required characteristics, morphology, and purity. As is known, sodium aluminate is produced by leaching bauxite at high temperatures and pressures. Under such conditions

and due to the heterogeneity of bauxite, decomposition occurs, not only of aluminum but also of certain impurities. Impurities have a detrimental effect on the quality of the product and reduce its price, leading to economic losses. In this paper, a three-stage purification strategy for removing impurities was investigated in order to obtain the purest possible product (alumina).

Several sets of experiments were performed, which were divided into three phases in which the effects of time, temperature, and concentration of lime were observed. The most important conclusion were as follows:

- The first stage of the process included the addition of lime in order to purify sodium aluminate from impurities of silicon and calcium. A detailed analysis of the results concluded that increasing the reaction time led to a decrease in the concentration of silicon compounds and an increase in the concentration of calcium compounds. An increase in temperature led to a decrease in silicon concentration and an increase in calcium concentration. Finally, a higher concentration of lime had a favorable effect on the reduction of Si, while the concentration of Ca remained constant (lower compared to the initial one). The optimal parameters for the first stage of the process were time of 60 min, temperature of 70 °C, and lime concentration of 20 g/L.
- In the second phase of the process, the focus was on removing impurities of iron, zinc, and copper, that is, their compounds were dissolved in sodium aluminate by adding specially produced aluminum hydroxide as a seed crystal. As for the influence of time on the purification of the solution from the given impurities, after just 45 min, the concentrations of copper and zinc dropped to 0, and the concentration of iron remained fairly constant and much lower than the initial one. By increasing the temperature, the concentrations of Fe and Zn impurities increased, so a lower temperature favored better cleaning. At a seed crystal concentration of 5 g/L, the proportion of impurities already dropped significantly; the optimal concentration value was 10 g/L.
- In the last stage, the crystallized solid phase purified from the mentioned impurities was subjected to hydrothermal washing in order to remove sodium oxide. In this case, by increasing all three parameters (time, temperature, and hydrate concentration), a cleaner product was obtained. The optimal values were time of 120 min, temperature of 210 °C, and concentration of 10 g/60 mL.

As already mentioned, it is very important to properly optimize the process conditions, which leads to a better purification effect with the lowest possible costs. It is necessary to sufficiently purify the solution with the lowest possible temperature of the solution as well as concentration in the shortest possible time interval.

**Author Contributions:** Conceptualization, V.D.; methodology, V.D.; validation, V.D., R.F. and Z.O.; formal analysis, M.P.; investigation, V.D.; resources, M.P.; data curation, writing—original draft preparation, V.D., D.K., S.S. (Slavko Smiljanic) and S.S. (Srecko Stopic). writing—review and editing, S.S. (Srecko Stopic); supervision, R.F., Z.O. and M.P. All authors have read and agreed to the published version of the manuscript.

**Funding:** This research received no external funding.

**Data Availability Statement:** Not applicable.

**Conflicts of Interest:** The authors declare no conflict of interest.

## Appendix A

The first purification phase (analysis of chemical solution before and after purification).

**Table A1.** Influence of time.

| | t | T | c | Al$_2$O$_3$ | NaO$_k$ | α$_k$ | Ca | Cr | Cu | Fe | Mg | Mn | Si | SiO$_2$ | Fe$_2$O$_3$ | ZnO | CaO | Al$_2$O$_3$ |
|---|---|---|---|---|---|---|---|---|---|---|---|---|---|---|---|---|---|---|
| | | | | | | | | | | | **Liquid Phase** | | | | **Solid Phase** | | | |
| | min | °C | gdm$^{-3}$ | gdm$^{-3}$ | gdm$^{-3}$ | | mgdm$^{-3}$ | mgdm$^{-3}$ | mgdm$^{-3}$ | mgdm$^{-3}$ | mgdm$^{-3}$ | mgdm$^{-3}$ | mgdm$^{-3}$ | % | % | % | % | % |
| | \multicolumn Initial aluminate solution | | | 171.87 | 156.55 | 1.50 | 11.00 | 0.04 | 6.60 | 16.96 | 2.08 | 0.16 | 43.6 | | | | | |
| 1A | 60 | 80 | 20 | 167.28 | 156.55 | 1.54 | 3.80 | - | 2.52 | 2.88 | 1.88 | - | 8.40 | 1.694 | 0.346 | 0.011 | 65.134 | 31.721 |
| 2A | 120 | 80 | 20 | 164.73 | 155.78 | 1.56 | 9.24 | - | 2.12 | 3.80 | 1.56 | - | 8.16 | 0.698 | 0.208 | 0.005 | 67.041 | 31.096 |
| 3A | 180 | 80 | 20 | 162.69 | 155.78 | 1.58 | 6.40 | - | 2.40 | 3.52 | 0.60 | - | 17.40 | 0.690 | 0.200 | 0.010 | 66.580 | 31.730 |
| 4A | 240 | 80 | 20 | 156.55 | 162.69 | 1.58 | 4.92 | - | 2.00 | 3.96 | 0.72 | - | 13.32 | 1.185 | 0.246 | 0.006 | 65.110 | 32.195 |
| 5A | 300 | 80 | 20 | 156.55 | 164.22 | 1.57 | 4.80 | - | 2.20 | 4.12 | 0.60 | - | 9.48 | 1.096 | 0.214 | 0.006 | 65.120 | 32.276 |

**Table A2.** Influence of temperature.

| | t | T | c | Al$_2$O$_3$ | NaO$_k$ | α$_k$ | Ca | Cr | Cu | Fe | Mg | Mn | Si | SiO$_2$ | Fe$_2$O$_3$ | ZnO | CaO | Al$_2$O$_3$ |
|---|---|---|---|---|---|---|---|---|---|---|---|---|---|---|---|---|---|---|
| | | | | | | | | | | | **Liquid Phase** | | | | **Solid Phase** | | | |
| | min | °C | gdm$^{-3}$ | gdm$^{-3}$ | gdm$^{-3}$ | | mgdm$^{-3}$ | mgdm$^{-3}$ | mgdm$^{-3}$ | mgdm$^{-3}$ | mgdm$^{-3}$ | mgdm$^{-3}$ | mgdm$^{-3}$ | % | % | % | % | % |
| | Initial aluminate solution | | | 170.34 | 156.55 | 1.51 | 10.64 | 0.04 | 6.76 | 18.56 | 0.56 | 0.12 | 43.88 | | | | | |
| 6A | 120 | 50 | 20 | 164.22 | 156.55 | 1.57 | 4.16 | - | 1.72 | 5.52 | 0.72 | - | 24.24 | 0.349 | 0.234 | 0.007 | 66.894 | 31.797 |
| 7A | 120 | 60 | 20 | 164.73 | 156.55 | 1.56 | 5.08 | - | 2.20 | 6.04 | 0.76 | - | 19.88 | 0.551 | 0.210 | 0.007 | 67.192 | 31.247 |
| 8A | 120 | 70 | 20 | 160.14 | 151.90 | 1.56 | 4.92 | - | 2.12 | 6.28 | 0.20 | - | 8.48 | 0.651 | 0.196 | 0.006 | 67.129 | 31.110 |
| 9A | 120 | 80 | 20 | 164.22 | 155.78 | 1.56 | 6.92 | - | 2.52 | 6.60 | 0.80 | - | 8.72 | 0.881 | 0.200 | 0.006 | 66.550 | 31.580 |

**Table A3.** Influence of lime concentration.

| | t | T | c | Al$_2$O$_3$ | NaO$_k$ | α$_k$ | Ca | Cr | Cu | Fe | Mg | Mn | Si | SiO$_2$ | Fe$_2$O$_3$ | ZnO | CaO | Al$_2$O$_3$ |
|---|---|---|---|---|---|---|---|---|---|---|---|---|---|---|---|---|---|---|
| | | | | | | | | | | | **Liquid Phase** | | | | **Solid Phase** | | | |
| | min | °C | gdm$^{-3}$ | gdm$^{-3}$ | gdm$^{-3}$ | | mgdm$^{-3}$ | mgdm$^{-3}$ | mgdm$^{-3}$ | mgdm$^{-3}$ | mgdm$^{-3}$ | mgdm$^{-3}$ | mgdm$^{-3}$ | % | % | % | % | % |
| | Initial aluminate solution | | | 168.30 | 155.00 | 1.51 | 11.76 | 0.04 | 6.80 | 17.28 | 0.44 | 0.08 | 51.52 | | | | | |
| 10A | 120 | 80 | 5 | 170.34 | 155.00 | 1.50 | 3.36 | o.d.l. | 3.00 | 9.16 | 0.16 | o.d.l. | 39.20 | 1.740 | 0.490 | 0.010 | 64.530 | 32.120 |
| 11A | 120 | 80 | 10 | 171.36 | 155.00 | 1.49 | 3.68 | o.d.l. | 3.12 | 6.60 | 0.20 | o.d.l. | 23.40 | 0.370 | 0.240 | 0.010 | 66.930 | 31.760 |
| 12A | 120 | 80 | 20 | 165.75 | 157.33 | 1.56 | 2.64 | o.d.l. | 1.08 | 2.24 | 0.08 | o.d.l. | 12.36 | 1.186 | 0.215 | 0.006 | 65.810 | 31.601 |
| 13A | 120 | 80 | 30 | 164.73 | 156.55 | 1.56 | 3.44 | o.d.l. | 1.20 | 2.24 | 0.16 | o.d.l. | 10.68 | 0.710 | 0.145 | 0.005 | 67.310 | 31.027 |
| 14A | 120 | 80 | 40 | 164.22 | 155.78 | 1.56 | 3.80 | o.d.l. | 0.96 | 4.20 | 0.20 | o.d.l. | 6.92 | 0.701 | 0.162 | 0.006 | 66.246 | 32.030 |

The second purification phase (analysis of chemical solution before and after purification).

**Table A4.** Influence of time.

| | t | T | c | NaO$_k$ | Al$_2$O$_3$ | α$_k$ | Ca | Cu | Fe | Mn | Zn | SiO$_2$ | Fe$_2$O$_3$ | Na$_2$O | ZnO | CaO |
|---|---|---|---|---|---|---|---|---|---|---|---|---|---|---|---|---|
| | | | | | | | | | | **Liquid Phase** | | | | **Solid Phase** | | |
| | min | °C | gdm$^{-3}$ | gdm$^{-3}$ | gdm$^{-3}$ | | mgdm$^{-3}$ | mgdm$^{-3}$ | mgdm$^{-3}$ | mgdm$^{-3}$ | mgdm$^{-3}$ | % | % | % | % | % |
| | Initial aluminate solution/hydrate | | | 155.78 | 168.30 | 1.52 | 9.60 | 6.44 | 22.32 | 0.08 | 9.32 | | | | | |
| 1B | 30 | 50 | 5 | 153.45 | 147.90 | 1.71 | 2.12 | 0.00 | 7.52 | 0.00 | 0.00 | 0.006 | 0.05 | 0.43 | 0.0305 | 0.018 |
| 2B | 45 | 50 | 5 | 153.45 | 142.80 | 1.77 | 2.44 | 0.00 | 7.96 | 0.00 | 0.00 | 0.005 | 0.04 | 0.48 | 0.0250 | 0.015 |
| 3B | 60 | 50 | 5 | 154.23 | 130.05 | 1.95 | 2.16 | 0.00 | 7.44 | 0.00 | 0.00 | 0.004 | 0.03 | 0.45 | 0.0185 | 0.013 |
| 4B | 120 | 50 | 5 | 150.00 | 113.22 | 2.25 | 2.28 | 0.00 | 8.04 | 0.00 | 0.16 | 0.004 | 0.02 | 0.41 | 0.0186 | 0.012 |
| 5B | 150 | 50 | 5 | 154.23 | 110.16 | 2.30 | 2.56 | 0.00 | 8.56 | 0.00 | 0.84 | 0.001 | 0.02 | 0.41 | 0.0134 | 0.009 |

**Table A5.** Influence of temperature.

| | t | T | c | NaO$_k$ | Al$_2$O$_3$ | α$_k$ | Ca | Cu | Fe | Mn | Zn | SiO$_2$ | Fe$_2$O$_3$ | Na$_2$O | ZnO | CaO |
|---|---|---|---|---|---|---|---|---|---|---|---|---|---|---|---|---|
| | | | | | | | | | | **Liquid Phase** | | | | **Solid Phase** | | |
| | min | °C | gdm$^{-3}$ | gdm$^{-3}$ | gdm$^{-3}$ | | mgdm$^{-3}$ | mgdm$^{-3}$ | mgdm$^{-3}$ | mgdm$^{-3}$ | mgdm$^{-3}$ | % | % | % | % | % |
| | Initial aluminate solution/hydrate | | | 155.00 | 166.26 | 1.53 | 9.60 | 6.72 | 20.00 | 0.08 | 9.44 | | | | | |
| 6B | 120 | 45 | 5 | 155.00 | 158.10 | 1.61 | 1.60 | 0.00 | 6.80 | 0.00 | 2.08 | 0.0050 | 0.12 | 0.57 | 0.0640 | 0.046 |
| 7B | 120 | 50 | 5 | 154.23 | 150.96 | 1.68 | 1.48 | 0.00 | 6.80 | 0.00 | 0.32 | 0.0007 | 0.06 | 0.68 | 0.0445 | 0.026 |
| 8B | 120 | 60 | 5 | 155.78 | 144.84 | 1.77 | 2.24 | 0.00 | 8.52 | 0.00 | 1.04 | 0.0020 | 0.05 | 0.35 | 0.0338 | 0.022 |
| 9B | 120 | 70 | 5 | 155.78 | 142.80 | 1.79 | 2.76 | 0.00 | 8.68 | 0.00 | 4.04 | 0.0050 | 0.059 | 0.13 | 0.0250 | 0.026 |

**Table A6.** Influence of concentration.

| | | | | Liquid Phase | | | | | | | | Solid Phase | | | | |
|---|---|---|---|---|---|---|---|---|---|---|---|---|---|---|---|---|
| | t | T | c | $NaO_k$ | $Al_2O_3$ | $\alpha_k$ | Ca | Cu | Fe | Mn | Zn | $SiO_2$ | $Fe_2O_3$ | $Na_2O$ | ZnO | CaO |
| | min | °C | gdm$^{-3}$ | gdm$^{-3}$ | gdm$^{-3}$ | | mgdm$^{-3}$ | mgdm$^{-3}$ | mgdm$^{-3}$ | mgdm$^{-3}$ | mgdm$^{-3}$ | % | % | % | % | % |
| Initial aluminate solution/hydrate | | | | 156.55 | 170.34 | 1.51 | 9.08 | 6.64 | 13.16 | 0.04 | 9.40 | | | | | |
| 10B | 120 | 50 | 5 | 156.55 | 156.57 | 1.64 | 1.48 | 0.00 | 0.00 | 0.00 | 1.76 | 0.005 | 0.082 | 0.58 | 0.0537 | 0.045 |
| 11B | 120 | 50 | 10 | 155.00 | 114.75 | 2.22 | 0.96 | 0.00 | 0.00 | 0.00 | 0.00 | 0.003 | 0.020 | 0.40 | 0.0153 | 0.011 |
| 12B | 120 | 50 | 20 | 150.35 | 99.45 | 2.19 | 1.40 | 0.00 | 0.52 | 0.00 | 0.00 | 0.005 | 0.020 | 0.34 | 0.0126 | 0.010 |
| 13B | 120 | 50 | 30 | 147.25 | 89.25 | 2.71 | 1.48 | 0.00 | 0.64 | 0.00 | 0.00 | 0.004 | 0.020 | 0.30 | 0.0119 | 0.009 |
| 14B | 120 | 50 | 40 | 137.95 | 82.88 | 2.74 | 1.08 | 0.00 | 0.60 | 0.00 | 0.00 | 0.004 | 0.020 | 0.28 | 0.0140 | 0.009 |

The third purification phase (analysis of solid residue after purification).

**Table A7.** Influence of time.

| | $SiO_2$ | $Fe_2O_3$ | Ignition Loss | $Na_2O$ | ZnO | CaO |
|---|---|---|---|---|---|---|
| | % | % | % | % | % | % |
| Initial hydrate | 0.0033 | 0.0096 | 34.24 | 0.06 | 0.0076 | 0.005 |
| 1C | 0.008 | 0.013 | 34.14 | 0.04 | 0.0058 | 0.006 |
| 2C | 0.007 | 0.012 | 30.27 | 0.03 | 0.006 | 0.004 |
| 3C | 0.008 | 0.013 | 21.42 | 0.004 | 0.0072 | 0.005 |
| 4C | 0.007 | 0.013 | 19.36 | 0 | 0.0078 | 0.005 |

**Table A8.** Influence of temperature.

| | $SiO_2$ | $Fe_2O_3$ | Ignition Loss | $Na_2O$ | ZnO | CaO |
|---|---|---|---|---|---|---|
| | % | % | % | % | % | % |
| Initial hydrate | 0.0033 | 0.0096 | 34.24 | 0.06 | 0.0076 | 0.005 |
| 5C | 0.003 | 0.01 | 33.64 | 0.04 | 0.006 | 0.004 |
| 6C | 0.003 | 0.011 | 32.71 | 0.03 | 0.0059 | 0.004 |
| 7C | 0.005 | 0.014 | 21.68 | 0.01 | 0.0069 | 0.005 |
| 8C | 0.004 | 0.015 | 19.99 | 0 | 0.0072 | 0.005 |
| 9C | 0.006 | 0.015 | 18.48 | 0 | 0.0075 | 0.005 |
| 10C | 0.006 | 0.014 | 18.8 | 0 | 0.007 | 0.005 |

**Table A9.** Influence of concentration of solid phase during washing.

| | $SiO_2$ | $Fe_2O_3$ | Ignition loss | $Na_2O$ | ZnO | CaO |
|---|---|---|---|---|---|---|
| | % | % | % | % | % | % |
| Initial hydrate | 0.0033 | 0.0096 | 34.24 | 0.06 | 0.0076 | 0.005 |
| 11C | 0.01 | 0.016 | 16.99 | 0.01 | 0.0075 | 0.007 |
| 12C | 0.008 | 0.015 | 17.04 | 0.01 | 0.0072 | 0.005 |
| 13C | 0.002 | 0.014 | 19.06 | 0 | 0.0071 | 0.005 |
| 14C | 0.001 | 0.013 | 20.2 | 0 | 0.0068 | 0.006 |
| 15C | 0.001 | 0.013 | 24.83 | 0 | 0.0075 | 0.005 |

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
