# Peer review of "Influence of Process Parameters in Three-Stage Purification of Aluminate Solution and Aluminum Hydroxide"

_metals, doi:10.3390/met13111816_

Round 1

Reviewer 1 Report

Comments and Suggestions for Authors

The present manuscript deals with the purification of the aluminate solution obtained during the Bayers process. The article addresses some innovative methods for removing impurities from the solution, although it is difficult for readers to understand the way it is presented. Please address the following comments:

Please correct the word on line 14 “Whait”?

The English of the manuscript requires revision.

Correct the typo error on line 127 “Crystalization od”. Similarly on line 128 “durin”.

Authors are suggested to remove Fig.2 and 3.

Shift the paragraph written on line 189 to 200, to the introduction section and rewrite the procedure section.

What is “Na2Oc”?

Authors are suggested to include all the reactions involved in the synthesis and purification processes of aluminate solution.

Report the complete chemical analysis and pH of synthetic aluminate solution.

Please report complete experimental conditions in the caption of each figure and include data points for other elements to understand the behavior of these metal ions present in the solution.

The data points for Ca and Si at 80oC in Figure 6 do not match the data outlined in Figure 5. In Figure 5 the initial concentrations of Ca and Si are ~43 and 11 mg/dm3 respectively while that of Ca and Si are 24 and 4 mg/dm3 respectively in Fig. 6. What is the actual initial concentration of these metal ions?

Authors are suggested to report the chemical analysis of the solution obtained after removal of Ca and Si.

Comments on the Quality of English Language

The English of the manuscript requires revision.

Author Response

Dear Reviewer,

thank you very much for your valuable comments and  invested time to improve our paper. Firstly we submitted our paper on 17 pages. After your evaluation  we will submit our results in our improved version on 22 Pages. Additionally we added  6 new citations in references, 2 SEM and 2 EDX -analysis (after I and II purificaton Phase); 2 XRD-Analysis (after I and III phase) und 9 Tables in APPENDIX (chemical composition of initial aluminate solution, chemical composition of solution after treatment, and content of solid phase) in order to explain better our work

According to your following comments we are sending our anwers:

Please correct the word on line 14 “Whait”?

We changed it (in red color)

The English of the manuscript requires revision.

It will be changed at the End of this work with an assistance of MDPI and our colleagues at the RWTH University. Personally, we have just improved our text with Checking of English.

Correct the typo error on line 127 “Crystalization od”. Similarly on line 128 “durin”.

We made it.

Authors are suggested to remove Fig.2 and 3.

We removed it.

Shift the paragraph written on line 189 to 200, to the introduction section and rewrite the procedure section.

We made it.

What is “Na2Oc”?

In Bayer technology sodium in solutions is shown as Na2O total= Na2O carbonate +Na2O caustic. Carbonate is sodium in the form of Na2CO3 and is practically inactive in the leaching process, while caustic Na2O is considered free and as such dissolves the aluminum found in bauxite. In our system, Na2Oc refers to Na2O bound in sodium aluminate and free sodium in the form of sodium hydroxide (Caustic soda - that's why c). The concentration of Na2O in caustic form is important for the state of aluminate, while the molar ratio of Na2O/Al2O3 expresses the module of aluminate, on which its stability and the conditions for crystallization of hydroxide directly depend.

Authors are suggested to include all the reactions involved in the synthesis and purification processes of aluminate solution.

We injected the chemical reactions for the first purification step.

Al2O3 + 2 NaOH = 2NaAlO2 + 2H2O      

??(??)3(?)+??O?(??)↔????(??)4(??)

SiO2 + 2NaOH = Na2SiO3+  H20

2NaAlO2 + Ca(OH)2= Ca3Al2O+ 2 NaOH    

3??(??)2+2?????2+???2???3+(4−?)?2?= =3???·??2?3·????2(6−2?)?2O+2(1+?)??Ö?

Na2SiO3 + Ca(OH)= CaAlSiO3 + 2 NaOH

3 Al2Si2O5(OH)4 + 18 NaOH →6Na2SiO3 + 6 NaAl(OH)4 + 3 H2

6 Na2SiO3 + 6 NaAl(OH)4 + NaX → Na6[Al6Si6O24]∙ 2 NaX + 12 NaOH + 6H2O

The purification of elements in second step with an addition of fine aluminium hydroxide is via an adsorption mechanism, and no chemical reaction. This is an innovative step of this work

Washing of sodium is in the third step, and no chemical reaction.

Report the complete chemical analysis and pH of synthetic aluminate solution.

pH-Value was high alkaline. We did not measured it, but it is alkaline because of an addition of lime  (expected between 12 and 14)

Please report complete experimental conditions in the caption of each figure and include data points for other elements to understand the behavior of these metal ions present in the solution.

We included it in the caption of each Figure and Additional  in Appendix we added 9 TAbles, according to our Diagrams.

The data points for Ca and Si at 80oC in Figure 6 do not match the data outlined in Figure 5. In Figure 5 the initial concentrations of Ca and Si are ~43 and 11 mg/dm3 respectively while that of Ca and Si are 24 and 4 mg/dm3 respectively in Fig. 6. What is the actual initial concentration of these metal ions?

The initial concentration of solution is partially written on diagrams and in our Tables in Appendix in detail.. Unfortunately, these studied solutions were not stable during time. Therefore you found some difference. We performed  our experiments in long period time with different solutions in frame of one Doctoral work and some industrial investigation in Company Alumina, Zvornik Bosnia and Herzegovina. Therefore we added 9 TAbles for 9 Diagrams with initial concentration of aluminate solution and final concentration after treatment. You can follow not only the concentration of Si and Ca, but also other impurities such as Cu, Mn, Mg.

Authors are suggested to report the chemical analysis of the solution obtained after removal of Ca and Si!

Chemical analysis of solution obtained after removal of Ca and Si is presented in Appendix in 3 different tables for first purification step (analysis of Liquid and solid Phase). Please to check it!

Thank you very much for your big assistance in this matter!

Reviewer 2 Report

Comments and Suggestions for Authors

The manuscript deals with decontamination of a simulated aluminate solution and its following products to obtain pure alumina. In my opinion, the manuscript would be interesting for the journal readers. Minor revision is needed, which mainly related to work structure.

I highlighted some recommendations, comments and questions for authors according to the manuscript:

Line 1-3. Title. In my opinion, the title is not very good because it doesn’t correspond to the content of the manuscript. In my opinion, it would be better to emphasize 3-stage purification.

Line 63-66. It would be better to highlight the subject of the paper in the last paragraph of the Introduction Section.

Line 110-115. It would be better to write these sentences without numbering.

Line 118-124. “weight percent” and “% by weight” can be replaced by “wt. %”

Line 130. “The obtaines alumina…” Typo.

Line 154-157. Please, add the composition of the simulated solution and the initial aluminum hydroxide for third stage including contents of the impurities.

Line 188-227. Section 2.4. This text should be involved in the Introduction.

Line 233-246. This text should be involved in the Section 2.4.

Line 295. Why didn't you do XRD analysis of the solid residue?

Line 304-312. This text should be involved in the Section 2.4. Please, clarify in the text, how did you prepare seed crystals?

Comments on the Quality of English Language

English is not good. Extensive editing is required.

Author Response

Dear Reviewer,

thank you very much for your valuable comments and  invested time to improve our paper. Firstly we submitted our paper on 17 pages. After your evaluation  we will submit our results in our improved version on 22 Pages. Additionally we added  6 new citations in references, 2 SEM and 2 EDX -analysis (after I and II purificaton Phase); 2 XRD-Analysis (after I and III phase) und 9 Tables in APPENDIX (chemical composition of initial aluminate solution, chemical composition of solution after treatment, and content of solid phase) in order to explain better our work

According to your recommendations, comments and questions for authors we are sending our Answers.

Line 1-3. Title. In my opinion, the title is not very good because it doesn’t correspond to the content of the manuscript. In my opinion, it would be better to emphasize 3-stage purification.

we changed  a title of this paper.

New title; Influence of process parameters in three-stage purification of aluminate solution

Line 63-66. It would be better to highlight the subject of the paper in the last paragraph of the Introduction Section.

we made it.  "

In this work, the focus is on the removal of impurities of iron, zinc, silicon, calcium and sodium. Some of these ways are removal of Zn and Cu by filtration through a layer of granules containing iron trioxide [10], removal of Zn by addition of ZnS germs in the presence of sulfide ion [13], removal of colloidal iron by filtration through a suitable polymer [13], and removal of Fe by filtration through layer of sand [9]. The main aim of the work is to remove the mentioned impurities from synthetic sodium aluminate, what is prepared from a non-metallurgical hydrate that dissolves in sodium hydroxide. Sodium hydroxide solution is prepared by dissolving granulated solid NaOH in a certain amount of water.

Line 110-115. It would be better to write these sentences without numbering.

We written these sentences without numbering.

Based on the results presented here, it can be firstly concluded that an increasing the time of contact between liquor and seed crystals has a positive effect on the removal of iron, zinc and copper liquor impurities. Than an increasing the temperature reduces the impurity removal efficiency from the sodium aluminate solution indirectly because with higher process temperature rate of precipitation is lower and solubility of all impurities is higher and at a temperature of 40 °C, good impurity removal results are achieved.

Line 118-124. “weight percent” and “% by weight” can be replaced by “wt. %”

We changed it in our text. "The particles preferably have an Fe2O3 content of about 40 to 100 wt. %. For more effective removal of zinc, the particles are coated with a metal sulfide, preferably zinc sulfide [15]. A maximum zinc content of 0.03 wt. % can be tolerated in certain alloys. Theoretically, a zinc content of 0.03 weight percent in the metal is derived from an aluminum oxide containing about 0.02 wt. % ZnO.

Line 130. “The obtaines alumina…” Typo.

we changed it.

Line 154-157. Please, add the composition of the simulated solution and the initial aluminum hydroxide for third stage including contents of the impurities.

We added chemical composition for all 3-stage purification (totally 9 tables with content of impurities.) in Appendix

Line 188-227. Section 2.4. This text should be involved in the Introduction.

We changed our abstract and introduction.

Line 233-246. This text should be involved in the Section 2.4.

We improved the Section 2.4 with new chemical equations:

Al2O3 + 2 NaOH = 2NaAlO2 + 2H2O      

??(??)3(?)+??O?(??)↔????(??)4(??)

SiO2 + 2NaOH = Na2SiO3+  H20

2NaAlO2 + Ca(OH)2= Ca3Al2O+ 2 NaOH    

3??(??)2+2?????2+???2???3+(4−?)?2?= =3???·??2?3·????2(6−2?)?2O+2(1+?)??Ö?

Na2SiO3 + Ca(OH)= CaAlSiO3 + 2 NaOH

3 Al2Si2O5(OH)4 + 18 NaOH →6Na2SiO3 + 6 NaAl(OH)4 + 3 H2

6 Na2SiO3 + 6 NaAl(OH)4 + NaX → Na6[Al6Si6O24]∙ 2 NaX + 12 NaOH + 6H2O

The purification of elements in second step with an addition of fine aluminium hydroxide is via an adsorption mechanism, and no chemical reaction. This is an innovative step of this work. Washing of sodium is in the third step, and no chemical reaction.

Line 295. Why didn't you do XRD analysis of the solid residue?

In this new version we injected two XRD-analysis of solid residue (after first purification step and the third purification step)

Line 304-312. This text should be involved in the Section 2.4. Please, clarify in the text, how did you prepare seed crystals?

The preparation of seed crystals was performed in company Alumina in Zvornik. They  have no permission to give all details. They used aluminium hidroxide, Alumina, Zvornik and prepared seed crystal after grinding of water suspension. 

Comments on the Quality of English Language

English is not good. Extensive editing is required.

It will be changed at the End of this work with an assistance of MDPI and our colleagues at the RWTH University. Personally, we have just improved our text with Checking of English.

Thank you very much for your big assistance in this evaluation

Reviewer 3 Report

Comments and Suggestions for Authors

The paper under review represents a nice technical contribution and in my opinion, it meets the publishing standards of “Metals” and it could be published after minor revision

Comments

1. Abstract

The first 10 lines should be transferred in the Introduction section

2. Na is removed by the process of hydrothermal washing of Al2O3 x 3H20

The correct is “Al2O3 x 3H2O”

3. the monitoring of the mentioned parameters (t, c, ? )

The parameters are not mentioned anywhere in the Abstract and should be defined

4. Figure 1. Structure of the aluminate ion.

Figure 1 is not mentioned in the Text.

5. in equilibrium with hemicarbonate species C4A

C4A: 4CaO·Al2O3.

6. Figure 2. Varian NexION 5000 ICP-MS instrument

Figure 2 should be removed

7. The chemical composition of the sample was determined by fluorescence X-ray struc-178 tural spectroscopy (EDX) on a Shimadzu 8000 device

Bulk analysis, especially for impurities, should be carried out by AAS or ICP, after dissolution

8.  Figure 3. a) Characterization device: Shimadzu 8000 b) Shimadzu IRAffinity

Figure 3 should be removed

9. Phase I, Phase II, Phase III.

XRD and SEM/EDS of the produced residues/solids after each stage would be very helpful

10. Phase II: Rapid crystallization (decomposition) by adding specially prepared aluminum hydroxide.

Why decomposition?

Also, what do the Authors mean with “specially prepared aluminum hydroxide”?

Author Response

Dear Reviewer,

thank you very much for your valuable comments and invested time, that helped us to improve our paper.

According to your comments, we improved our text!

  1. Abstract

The first 10 lines should be transferred in the Introduction section

We made it!

  1. Na is removed by the process of hydrothermal washing of Al2O3 x 3H20

The correct is “Al2O3 x 3H2O”. We changed it!

  1. the monitoring of the mentioned parameters (t, c, ? )

The parameters are not mentioned anywhere in the Abstract and should be defined

We improved it!

"The focus of this work is the monitoring of the mentioned parameters such as temperature (T), reaction time (t) and concentration  (c) in order to remove the mentioned impurities to obtain the purest possible product that would be an adequate precursor for special types of aluminas.

  1. Figure 1. Structure of the aluminate ion.

Figure 1 is not mentioned in the Text.

 We added:

The structure of aluminate ion is shown at the Figure 1.

  1. in equilibrium with hemicarbonate species C4A

C4A: 4CaO·Al2O3.

We changed it.

  1. Figure 2. Varian NexION 5000 ICP-MS instrument

Figure 2 should be removed

We removed it.

  1. The chemical composition of the sample was determined by fluorescence X-ray struc-178 tural spectroscopy (EDX) on a Shimadzu 8000 device

Bulk analysis, especially for impurities, should be carried out by AAS or ICP, after dissolution

Bulk analysis, especially for impurities, was carried out by AAS or ICP, after dissolution

  1. Figure 3. a) Characterization device: Shimadzu 8000 b) Shimadzu IRAffinity

Figure 3 should be removed

We removed it!

  1. Phase I, Phase II, Phase III.

XRD and SEM/EDS of the produced residues/solids after each stage would be very helpful

We added SEM/EDS analysis of the produced residues/solids in purification stage 1 and 2, and XRD analysis for the stage 3.

  1. Phase II: Rapid crystallization (decomposition) by adding specially prepared aluminum hydroxide.

Why decomposition?

Also, what do the Authors mean with “specially prepared aluminum hydroxide”?

We changed it. Rapid crystallization  was performed by adding alluminium hydroxide.

Reviewer 4 Report

Comments and Suggestions for Authors

This work examined the effects of the purification process parameters for reducing impurities on the extraction of alumina from bauxite ore. The authors divided the purification process into three phases: phase I removed Ca and Si; phase II removed Zn, Fe and Cu; and phase III removed Na. The authors successfully reduced these impurities by varying the conditions such as time, temperature and concentration.

However, the information revealed in this paper appears to be limited.  This paper has the following fatal weaknesses, which will not be overcome in the review process. The reviewer recommends Reject for this paper.

1. The authors examined the effects of the process parameters at each purification stage. However, little discussion was given to the chemistry behind the results. The reviewer cannot understand why the authors consider the addition of lime to be effective in reducing Ca.

The contribution of this study to the development of this research area is considered rather limited.

2. Simple but serious errors, and ambiguities are found throughout, hindering understanding of the results.

For example,

(1) "g cm3" in "0.0494 g dm3 (P. 3, L. 133)" is not a unit of concentration.

(2) Instead of describing the impurities by the elemental symbols (Fe, Zn, Cu, Ca, Na, and Si), specify the chemical forms, such as Zn2+, Cu2+, etc.

(3) The Abstract should briefly summarize the main points of this work. General background (L. 13-25) is provided in the Introduction.

(4) The photographs of the analytical instruments are not necessary (Fig. 2 and 3). 

Well-known general-information about the analytical techniques may be omitted if the features are not directly used in this work. For example, "It is known and used for the detection of metals and several non-metals in liquid samples at very low concentrations (ppm-range). It can detect different isotopes of the same element, making it a versatile isotopic labeling tool. Compared to atomic absorption spectroscopy (AAS), ICP has greater speed, precision and sensitivity". (P. 4, L. 160-164)

Comments on the Quality of English Language

English editing to check spelling and grammar is required.

For example,

"The obtaines alumina (Alumina I) is …" should be "The obtained alumina (Alumina I) is …" (P. 3, L. 130)

"3H20" <- It should be H2"O", not H2"zero". (P. 1, L. 29) 

"from Aluminates" <- Why does "aluminates" start with a capital letter? (P. 2, L. 48)

Author Response

Dear Reviewer,

thank you very much for your valuable comments and  invested time to improve our paper. Firstly we submitted our paper on 17 pages. After your evaluation  we will submit our results in our improved version on 22 Pages. Additionally we added  6 new citation in references, 2 SEM and 2 EDX -analysis (after I and II purificaton Phase); 2 XRD-Analysis (after I and III phase) und 9 Tables in APPENDIX (chemical composition of initial aluminate solution, chemical composition of solution after treatment, and content of solid phase) in order to explain better our work

According to your following comments we are sending our anwers

However, the information revealed in this paper appears to be limited.  This paper has the following fatal weaknesses, which will not be overcome in the review process. The reviewer recommends Reject for this paper.

We can not understand  what are "fatal weekness" in this research. We found 5 different publications in last 4 years, that are studied same problem.

  1. Steven J. Healy, Bayer Process Impurity and Their Management, 375-422 (chapter in Book; Smelter Grade Alumina from Bauxite History, Best Practices, and Future Challenges, Springer Series in Materials Science 320, Benny E. Raahauge and Fred S. Williams (Editors), Springer,
  2. Rossetto, D; M , Rodrigues, D , LaMacchia, R. Optimization of Lime Dosage to Digestion at Alunorte Refinery Using Monte Carlo Methods, Travaux 46, Proceedings of 35th International ICSOBA Conference, Hamburg, Germany, 2 – 5 October, 2017., 205-207.
  3. Ye,,. Feng, P., Lu, J., Zhao, L., Liu, Q., Zhang, Q., Liu, J, Bullard, J., Solubility of tricalcium sulfate from 10 to 40°C, Cement and concrete research, Volume 162, December 2022, 106989
  4. Ye, S.,Feng, P., Liu, Y,, Liu, J., Bullard, J. In situ nano-scale observation of C3A dissolution in water, Cement and concrete Research, 2020, 132, 10644
  5. Arikan, H, Demir, G-, Vural, S., Investigation of lime usage impacts on bauxite Procesiblity at ETI Aluminiym Plsnt, Industrial Journal of Industrial Chemistry,, 2029, 10, 57-66, 10.1007/s40090-019-0171-x

We injected the chemical reactions for the first purification step.

Al2O3 + 2 NaOH = 2NaAlO2 + 2H2O      

??(??)3(?)+??O?(??)↔????(??)4(??)

SiO2 + 2NaOH = Na2SiO3+  H20

2NaAlO2 + Ca(OH)2= Ca3Al2O+ 2 NaOH    

3??(??)2+2?????2+???2??O3+(4−?)?2?= =3???·??2?3·????2(6−2?)?2O+2(1+?)??O?

Na2SiO3 + Ca(OH)= CaAlSiO3 + 2 NaOH

3 Al2Si2O5(OH)4 + 18 NaOH →6Na2SiO3 + 6 NaAl(OH)4 + 3 H2

6 Na2SiO3 + 6 NaAl(OH)4 + NaX → Na6[Al6Si6O24]∙ 2 NaX + 12 NaOH + 6H2O

The purification of elements in second step with an addition of fine aluminium hydroxide is via an adsorption mechanism, and no chemical reaction. This is an innovative step of this work

Washing of sodium is in the third step, and no chemical reaction.

The reviewer cannot understand why the authors consider the addition of lime to be effective in reducing Ca.

You have right. An addition of lime is not directly applied for a reducing of Ca from Solution. In our case it can be driving force to move available calcium to be used for purification of solution with an added calcium.

Generally, Lime usage 
1. Enhances the boehmite and diaspore dissolution rates. It can give an opportunity to process higher diaspore content bauxites in existing plants.
2. Provides a signifcant decrease of Na2O/SiO2 ratio in red mud, which means up to 40% caustic savings can be achieved.
3. Gives higher settling rates of red mud. It will assist in obtaining a more stable operation of the settler and washer circuit which can result in lower coagulant and focculant consumptions.
4. Provides about 5% saving at specific steam consumption.

Additionally, the impurity level of the process liquor can be controlled with lime In addition, resulting in 5–15% alumina product quality improvement, what is confirmed by ETI Aluminyum A.Ş., the primary aluminium manufacturer of Turkey (Ref.5). 

Simple but serious errors, and ambiguities are found throughout, hindering understanding of the results.

For example,

      • "g cm3" in "0.0494 g dm3 (P. 3, L. 133)" is not a unit of concentration.

We changed it.

      • Instead of describing the impurities by the elemental symbols (Fe, Zn, Cu, Ca, Na, and Si), specify the chemical forms, such as Zn2+, Cu2+, etc.

We changed it.

      • The Abstract should briefly summarize the main points of this work. General background (L. 13-25) is provided in the Introduction.

We improved our introduction and abstract.

      • The photographs of the analytical instruments are not necessary (Fig. 2 and 3). 

We removed Photographs 2 and 3.

  1. We added well-known general-information about the analytical techniques may be omitted if the features are not directly used in this work. For example, "It is known and used for the detection of metals and several non-metals in liquid samples at very low concentrations (ppm-range). It can detect different isotopes of the same element, making it a versatile isotopic labeling tool. Compared to atomic absorption spectroscopy (AAS), ICP has greater speed, precision and sensitivity". (P. 4, L. 160-164)

We changed it.

Comments on the Quality of English Language

English editing to check spelling and grammar is required.

For example,

"The obtaines alumina (Alumina I) is …" should be "The obtained alumina (Alumina I) is …" (P. 3, L. 130)

We changed it

"3H20" <- It should be H2"O", not H2"zero". (P. 1, L. 29) 

We changed it

"from Aluminates" <- Why does "aluminates" start with a capital letter? (P. 2, L. 48)

It will be changed at the End of this work with an assistance of MDPI and our colleagues at the RWTH University. Personally, we have just improved our text with Checking of English.

Thank you very much for your support.

Round 2

Reviewer 1 Report

Comments and Suggestions for Authors

Remove the Fig.1.

Please report what is " non-metallurgical hydrate" in the procedure. Move the lines from 251-261 into the procedure section.

The authors are advised to move equation 1 to equation 6 in section 3.1.

In what form lime has been introduced to the system, solid or solution? 

Please re-write the first paragraph of section 3.2. "what is one innovative step of this work, what is missing in literature based on purification of solution"?

The chemical composition of the initial solution should be reported in the main text instead of an appendix. Similarly include the chemical composition of the solution obtained after each stage of purification (at the optimum condition) in the main text and include them in the schematic diagram as well. Because the manuscript discusses the purification of the solution, the main text must include the composition of the parent solution used at every stage.

Fig. 4 " Change in C". Please correct.

Delete tables 1, 2 and 3.

Please present the complete process flowsheet at the end of the results and discussion.

Comments on the Quality of English Language

OK.

Author Response

Dear Reviewer,

thank you very much for your invested time and valuable comments. Attached to your questions we are sending our answers.

Remove the Fig.1.

It is easy to remove our Figure 1, but we found that is better for readers to have Figure 1 in Text in order to better understand the structure of the aluminate ion. Therefore, if you agreed, we will not remove the Figure 1.

Please report what is " non-metallurgical hydrate" in the procedure. Move the lines from 251-261 into the procedure section.

Non-metallurgical hydrate is aluminum hydroxide from the production process that is used as a chemical grade for applications that do not involve the production of alumina for metallic aluminum. We added it in our text.

The authors are advised to move equation 1 to equation 6 in section 3.1.

Thank you for your comment. We moved equation 1 to equation 6 in section 3.1.

In what form lime has been introduced to the system, solid or solution? 

Lime is added as CaO, but immediately after the addition, reactions of lime "quenching" and the formation of Ca(OH)2 follow, We added it in text.

Please re-write the first paragraph of section 3.2. "what is one innovative step of this work, what is missing in literature based on purification of solution"?

We added new text

"Our new innovative strategy in this work is the multi-step cleaning of the aluminate solution with special reference to the removal of impurities from the second stage. It is known from the literature that the kinetics of sodium aluminosilicate (DSP) formation as well as crystallinity increase with increasing temperature, as shown at

6 Na2SiO3(aq) + 6 NaAl(OH)4(aq) + Na2X(aq) → Na6(Al6Si6O24) ∙ Na2X ∙3 H2O(s) + 12 NaOH (aq) + 3 H2O (aq)

X= ½ CO32-, ½ SO42-, OH-, Cl-

This process is most effective at temperatures of 90-100°C, however high temperatures were not favorable due to the increase of calcium in the solution. Due to the presence of calcium in excess, there is an exchange of calcium ions with sodium at the cation position in the aluminosilicate structure, reducing the loss of NaOH due to the formation of calcium aluminosilicate cancrinite. This is another mechanism for extracting calcium from the solution. Also, considering the cage structure of DSP (sodalite), which is also shown by the previous equation, there is the inclusion of carbonate and sulfate ions. Considering the addition of lime and the resulting reaction, it is very likely that there is an inclusion of calcium carbonate."

The chemical composition of the initial solution should be reported in the main text instead of an appendix. Similarly include the chemical composition of the solution obtained after each stage of purification (at the optimum condition) in the main text and include them in the schematic diagram as well. Because the manuscript discusses the purification of the solution, the main text must include the composition of the parent solution used at every stage.

You have right, but the complete picture about the chemical composition of the initial solution and the chemical composition of the solution obtained after each stage of purification is presented in APPENDIX  on two pages, that our readers have better review of our obtained results.  We are not sure that this movement can lead to some improvement in our text. Therefore we will not change it.

Fig. 4 " Change in C". Please correct.

We changed C  (now is Ca)

Delete tables 1, 2 and 3.

It is easy to delete Tables 1, 2 and 3. But we would like to point out our experimental work and our performed experiments with reactions parameters. Without our tables we have weakness in our paper. I am sorry, but we can not accept this suggestion.

Please present the complete process flowsheet at the end of the results and discussion.

Thank you for your comment, but complete process flowsheet as our research strategy belongs to our studied procedure (Chapter 2.4). We are not sure that it can lead for some improvement in our text. We can not change it.

Finally, Thank you very much for your comments that helped us to improve our text in order to be published in Metals.

Reviewer 4 Report

Comments and Suggestions for Authors

A weakness of this work was the lack of a chemistry-based discussion to explain the results of the process. The listing of chemical reaction formulae (P. 6, L. 224-229) eliminated this weakness somewhat, but the reviewer considers that further revisions are necessary before the paper can be accepted.

1. Please discuss the effects of the process parameters based on chemical considerations. For example, in P. 8, L. 294-296, the authors explain the effect of the Ca2+ concentration as "By increasing the concentration of calcium present, there is a more intense reaction with aluminum from the aluminate, which results in the formation of insoluble tricalcium aluminate". The phenomenon that adding lime (CaO) reduces Ca concentration sounds somewhat contradictory. The reason for this should be explained quantitatively on the basis of the pH of the solution or chemical equilibrium or something.

The authors show the temperature effect for Phase I in Fig. 4, where there is a clear decrease in the Si concentration at 70 deg. C. Please discuss what is happing in the solution near this temperature, based on chemical equilibrium or something else.

2. In equations (1)-(6), please specify which are the aqueous species to be detected as impurities and which are precipitates to be removed. (P. 6, L. 224-229).

3. The reviewer considers that silicon does not dissolve in the Si4+ form (P. 2, L. 72). Please provide the form of Si in aqueous solution assumed in this work.

4. It is difficult to identify the crystalline phases in the legend in Fig. 9 due to the use of similar colors and small characters with poor resolution. The identification of the crystalline phases should be shown more clearly.

 5. Please proofread the manuscript carefully again.

- The reactants and products in chemical equation (4) do not match. (P. 6, L. 227)

- Fig. 4 is captioned "Change in C and, Si concentrations….". The reviewer thinks that it should be "Change in Ca and Si concentrations….".

- The use of apparently different concentration units, "mg/L (in text)" and "mg/dm3 (in Fig. 4)", will confuse the readers, even though they are substantially the same.

Author Response

Dear Reviewer,

thank you very much for your valuable comments and invested time. We are sending our Answers to your comments.

Please discuss the effects of the process parameters based on chemical considerations. For example, in P. 8, L. 294-296, the authors explain the effect of the Ca2+ concentration as "By increasing the concentration of calcium present, there is a more intense reaction with aluminum from the aluminate, which results in the formation of insoluble tricalcium aluminate". The phenomenon that adding lime (CaO) reduces Ca concentration sounds somewhat contradictory. The reason for this should be explained quantitatively on the basis of the pH of the solution or chemical equilibrium or something.

By increasing the concentration of calcium present, there is a more intense reaction with aluminum from the aluminate, which results in the formation of insoluble tricalcium aluminate. The assumption is that this phenomenon occurs by two possible mechanisms.

One is that as a result of oversaturation of the solution with CaO oxide, a very fast reaction of formation of highly soluble tetracalcium aluminate occurs, which eventually transforms into hardly soluble tricalcium aluminate, which precipitates very quickly and is also the center of crystallization. At the same time, the synthesis of TCA is additionally accelerated, which ultimately leads to a decrease in the solubility of calcium in the given conditions (the given conditions favor the precipitation of TCA below the solubility limit of CaO). We suggested that the transition compound tetracalcium aluminate, which is highly soluble, pulls all the calcium out of the solution in the synthesis, and then there is a transformation and precipitation of TCA and CaCO3 from the solution.

Another mechanism that we assumed takes place is the adsorption process. According to the methods used, soluble calcium means the entire amount of Ca that is detected via ICP OES in the solution after filtering suspended particles on a laboratory filter. However, on these filters it is not possible to separate colloidal particles of calcium compounds that end up in the solution. Tetracalcium aluminate as an intermediate compound during transformation into TCA forms colloidal particles that are adsorbed on Ca(OH)2 particles, which also has limited solubility in sodium aluminate solution. In this way, by the mechanism of adsorption of nanosized particles on the formed crystals of TCA and insoluble Ca(OH)2, it is possible to separate them from the solution.

The authors show the temperature effect for Phase I in Fig. 4, where there is a clear decrease in the Si concentration at 70 deg. C. Please discuss what is happing in the solution near this temperature, based on chemical equilibrium or something else.

It is known from the literature that the kinetics of sodium aluminosilicate (DSP) formation as well as crystallinity increase with increasing temperature, as shown at

6Na2SiO3(aq) + 6NaAl(OH)4(aq) +Na2X(aq) → Na6(Al6Si6O24) ∙ Na2X ∙3H2O(s) + 12NaOH(aq) + 3H2O(aq)

X= ½CO32-, ½ SO42-, OH -, Cl-

This process is most effective at temperatures of 90-100°C, however high temperatures were not favorable due to the increase of calcium in the solution. Due to the presence of calcium in excess, there is an exchange of calcium ions with sodium at the cation position in the aluminosilicate structure, reducing the loss of NaOH due to the formation of calcium aluminosilicate cancrinite. This is another mechanism for extracting calcium from the solution. Also, considering the cage structure of DSP (sodalite), which is also shown by the previous equation, there is the inclusion of carbonate and sulfate ions. Considering the addition of lime and the resulting reaction, it is very likely that there is an inclusion of calcium carbonate.

 In equations (1)-(6), please specify which are the aqueous species to be detected as impurities and which are precipitates to be removed. (P. 6, L. 224-229)

Al2O3 + 2 NaOH → 2NaAlO2 (aq)  + 2H2O                                                                                                                       (1)

SiO2 + 2NaOH → Na2SiO3 (aq)  +  H2O                                                                                                               (2)

2NaAlO2 + Ca(OH)2 →Ca3Al2O6  (s) ↓+ 2 NaOH                                                                                                                               (3)

Na2AlSiO3 + Ca(OH)2 → CaAlSiO3 (s) ↓+ 2 NaOH                                                                                                                           (4)

3 Al2Si2O5(OH)4 (s) + 18NaOH →6Na2SiO3 + 6 NaAl(OH)4 + 3H2O                                                                    (5)

6 Na2SiO3 + 6NaAl(OH)4 + NaX → Na6[Al6Si6O24]∙2 NaX (s) ↓+ 12 NaOH + 6 H2O                                             (6)

3. The reviewer considers that silicon does not dissolve in the Si4+ form (P. 2, L. 72). Please provide the form of Si in aqueous solution assumed in this work.

The silicon comes from the aluminum hydroxide that was dissolved during the preparation of the synthetic solution. The consequence of the presence of silicon on the hydroxide is due to adsorption from the solution during the precipitation of Al(OH)3. It is dissolved in the form of sodium silicate Na2SiO3, which is formed when dissolved in NaOH solution

 It is difficult to identify the crystalline phases in the legend in Fig. 9 due to the use of similar colors and small characters with poor resolution. The identification of the crystalline phases should be shown more clearly.

Unfortunately, we can not offer the best resolution. We can only to put new table with list of these detected minerals. But we explained in text the presence of detected minerals, what is enough for our readers

Please proofread the manuscript carefully again.

The reactants and products in chemical equation (4) do not match. (P. 6, L. 227)

Thank you for your comment in order to change our mistake! We changed it

Na2AlSiO3 + Ca(OH)2 → CaAlSiO3 (s) ↓+ 2 NaOH

Fig. 4 is captioned "Change in C and, Si concentrations….". The reviewer thinks that it should be "Change in Ca and Si concentrations….".

Thank you for your comments. we changed it and have written

"Change in Ca and Si concentrations…

The use of apparently different concentration units, "mg/L (in text)" and "mg/dm3 (in Fig. 4)", will confuse the readers, even though they are substantially the same.

Thank you for your comments! We changed it in same units (g/L) in our text and improved it on y-axis on diagrams No. 3, 4, 5 and 10, 11, and 12. Now we prevented any confusion for our readers,

Finally, thank you for your excellent comments, which helped us to better explain our work and results.

Round 3

Reviewer 4 Report

Comments and Suggestions for Authors

The review had expected that the authors to take sufficient time to carefully revise the manuscript in order to complete the paper without any discrepancies and ambiguities. The reviewer is somewhat disappointed that the authors appear to be rushing to finish a job in a hurry.

L. 72 on P. 2 describes silicon ion as Si4+ at P. 2, L72, while L. 77-78 explains that the Si component is dissolved in the form of sodium silicate (it should more accurately be described as silicate ion (SiO3)2-).

The explanation of the effect of temperature on P. 3, L119-122 is very qualitative and no numerical evidence, such as solubility data for the substances involved, is provided.

Indicating the components by elemental symbols, such as Zn, Cu, and Fe, is very confusing, as they appear to exist in metallic form. They should be shown in their dissolved forms in solution.

"+" in "Na+" is not superscripted (P. 2, L72).

If X = 1/2 (CO3)2-, 1/2 (SO4)2-, OH-, Cl-, then the charges do not match in Na2X in P. 8, L307.

The phase identification in Fig. 9 is still difficult to see.

Why is the charge "3+" of Al written in "Ca3Al2 3+(OH)12" in P. 11, L. 390?

There is no definition for "TCA".

The authors are responsible for revising all inadequacies, including those that were not pointed out. If the authors do not do this, the paper must be rejected.

Author Response

Dear Reviewer,

thank you very much for your invested time and valuable comments. We are trying to give all answers regarding your comments in order to improve our text. My apology that we consumed your free time for new comments!

The review had expected that the authors to take sufficient time to carefully revise the manuscript in order to complete the paper without any discrepancies and ambiguities. The reviewer is somewhat disappointed that the authors appear to be rushing to finish a job in a hurry.

You have right that the authors appear to be rushing to finish a job in a hurry. But we have to respect this deadline by Metals (3 days). This time we have obtained again 3 days to finish it. From other side this is part of Doctoral Work of Damjanovic Vladimir. Together with Vladimir we as his supervisors are high motivated to offer our  results as soon as possible in order to improve our paper according to your comments.

We are sending our answers to your comments and questions.

L. 72 on P. 2 describes silicon ion as Si4+ at P. 2, L72, while L. 77-78 explains that the Si component is dissolved in the form of sodium silicate (it should more accurately be described as silicate ion (SiO3)2-).

We changed it in green color

The explanation of the effect of temperature on P. 3, L119-122 is very qualitative and no numerical evidence, such as solubility data for the substances involved, is provided.

I as metallurgist think that the essence here is not in the solubility limit of the mentioned impurities in the aluminate solution. The given conclusions are based on the assumption that the removal of impurities takes place by the mechanism of adsorption on the precipitated hydroxide. Consequently, the faster the precipitation, the more needle-shaped the newly formed aluminum hydroxide crystals are and the larger the specific surface area, and therefore favors more efficient adsorption. A lower temperature accelerates the precipitation process, while at higher temperatures the aluminate solution is more stable and therefore at higher temperatures the crystallization process is slower, directing the crystallization towards particles of a more regular shape and a lower specific surface area. Therefore, the solubility limits of the given compounds were not considered.

Indicating the components by elemental symbols, such as Zn, Cu, and Fe, is very confusing, as they appear to exist in metallic form. They should be shown in their dissolved forms in solution.

We changed it in green color!

"+" in "Na+" is not superscripted (P. 2, L72).

We changed it in green color!

Na+

If X = 1/2 (CO3)2-, 1/2 (SO4)2-, OH-, Cl-, then the charges do not match in Na2X in P. 8, L307.

We changed it in green color!

X= CO32-, SO42-, 2OH-, 2Cl-

If you accept our paper, additionally, after this round we will continue to work with Technical Assistant to prepare improved final proof and check our text again.

The phase identification in Fig. 9 is still difficult to see.

You have right! This is our problem. Therefore we worked on the improvement of this diagram! Now If we use a magnification (scale) of 121 % in text of our file, you can see all present phases in our diagram. Please, Can you check it?I am sure, that our readers can see it! Below our Fig. No.9,  we explained the presence of all phases in our text.

On the other side, I am trying to obtain a XRD- measurement file by Special XRD-Department in order to use special software to prepare new diagram in order to see it better. It takes time!

Why is the charge "3+" of Al written in "Ca3Al2 3+(OH)12" in P. 11, L. 390?

We changed it  in green color!

(Ca3Al2(OH)12). 

There is no definition for "TCA".

We improved it "tricalcium-aluminate 3CaO·Al2O3 (TCA)". 

The authors are responsible for revising all inadequacies, including those that were not pointed out. If the authors do not do this, the paper must be rejected.

Additionally we included the changes (in green color)  in new version regarding to the comment of our academic editor and obtained an improved version. We added new paper in references:

Vaughan, J., Peng, H., Seneviratne, D., Hodge, H., Hawker, W., Hayes, P., Staker, W., The Sandy Desilication Product Process Concept, Aluminum: Recycling and Environmental Footprint, Volume 71. No.9, 2019

Finally, your comments improved our version. Thank you very much for your work! We are very happy that we have obtained one improved paper, that we can include it in writing of Doctoral Work of Vladimir Damjanovic.

Round 4

Reviewer 4 Report

Comments and Suggestions for Authors

The reviewer thinks that the authors have responded sincerely with the best they can do at this time. 

The reviewer does not have any further comments on this paper.

Author Response

Dear Reviewer,

thank you very much for your excellent work and high support for our work. Your comment confirmed your high reputation in this field and helped us to improve our paper. You were our high motivation to prepare our improved paper